# Genetic variant effects on gene expression in human pancreatic islets and their implications for T2D

Ana Viñuela ⓘ et al.#

Most signals detected by genome-wide association studies map to non-coding sequence and their tissue-specific effects influence transcriptional regulation. However, key tissues and cell-types required for functional inference are absent from large-scale resources. Here we explore the relationship between genetic variants influencing predisposition to type 2 diabetes (T2D) and related glycemic traits, and human pancreatic islet transcription using data from 420 donors. We find: (a) 7741 *cis*-eQTLs in islets with a replication rate across 44 GTEx tissues between 40% and 73%; (b) marked overlap between islet *cis*-eQTL signals and active regulatory sequences in islets, with reduced eQTL effect size observed in the stretch enhancers most strongly implicated in GWAS signal location; (c) enrichment of islet *cis*-eQTL signals with T2D risk variants identified in genome-wide association studies; and (d) colocalization between 47 islet *cis*-eQTLs and variants influencing T2D or glycemic traits, including *DGKB* and *TCF7L2*. Our findings illustrate the advantages of performing functional and regulatory studies in disease relevant tissues.

#A list of authors and their affiliations appears at the end of the paper.

Genome-wide association studies (GWAS) have generated a growing inventory of genomic regions influencing type 2 diabetes (T2D) predisposition and related glycemic traits[1–3]. However, progress in defining the mechanisms whereby these associated variants mediate their impact on disease risk has been slow[4]. Over 90% of the associated signals map to noncoding sequence[5,6] complicating efforts to connect T2D-associated variants, with the transcripts and networks through which they exert their effects. One approach for addressing this variant-to-function challenge is to use expression quantitative trait loci (eQTL) mapping to characterize the impact of disease-associated regulatory variants on the expression of nearby genes[7].

Demonstrating that a disease-risk variant colocalizes with a *cis*-eQTL signal is consistent with a causal role for the transcript concerned, a hypothesis that can then be subject to more direct evaluation, for example, by perturbing the gene in suitable cellular or animal models. However, eQTL signals are often tissue specific[8]: consequently, the power to detect mechanistically informative expression effects is dependent on assaying expression data from sufficient numbers of samples across the range of disease-relevant tissues[7].

The pathogenesis of T2D involves dysfunction across multiple tissues, most obviously pancreatic islets, adipose, muscle, and liver. Risk variants that influence T2D predisposition through processes active in each of these have been reported (e.g., *KLF14* in adipose[9], *TBC1D4* in muscle[10], *ADCY5* in islets[11], and *GCKR* in liver[12]). However, multiple physiological and genomic analyses consistently indicate that islet dysfunction makes the greatest contribution to T2D risk[5,13,14]. Research access to human pancreatic islet material is therefore essential, and previous studies have demonstrated the potential of islet expression information to characterize T2D effector genes, such as *MTNR1B* and *ADCY5* (refs. [15–17]). However, access to human islet material is limited, and the largest published human islet RNA sequencing (RNA-Seq) dataset includes only 118 samples[17].

We constituted the InsPIRE (Integrated Network for Systematic analysis of Pancreatic Islet RNA Expression) consortium as a vehicle for the aggregation and joint analysis of human islet RNA-Seq data[15–18]. Here, we report analyses of 420 human islet preparations that provide a detailed landscape of the genetic regulation of gene expression in this key tissue, and its relationship to mechanisms of T2D predisposition.

Our research addresses questions with relevance beyond T2D. When a disease-relevant tissue is missing from reference datasets, such as GTEx, what additional value accrues from dedicated expression profiling from that missing tissue? What is the impact of tissue heterogeneity on the interpretation of eQTL data? What does the synthesis of tissue-specific epigenomic and expression data tell us about the coordination of upstream transcription factor (TF) regulators of gene expression? And, finally, what information do tissue-specific eQTL analyses provide about the regulatory mechanisms mediating disease predisposition?

## Results

**Characterization of genetic regulation of gene expression in islets**. We combined islet RNA-Seq with dense genome-wide genotype data from 420 individuals. Data from 196 of these individuals have been reported previously[15–18]. We aggregated, and then jointly mapped and reprocessed, all samples (median sequence-depth per sample ~60 M reads) to generate exon- and gene-level quantifications, using principal component (PC) methods to correct for technical and batch variation ("Methods" section; Supplementary Fig. 1).

To characterize the regulation of gene expression for the 17,914 protein-coding and long noncoding RNAs genes with quantifiable expression in these samples, we performed eQTL analysis (fastQTL[19]) on both exon- and gene-level expression measures, using all 8.05 M variants that passed quality control (QC; "Methods" section; Supplementary Data 1–4). In the gene-level analysis, we identified 4311 genes (eGenes) with significant *cis*-eQTLs at the gene level (FDR < 1%; *cis* defined as within 1 Mb of the transcription start site (TSS)). In the complementary exon-level analysis, which should enable a broader range of transcriptional effects to be captured, particularly those involving splicing, we detected 6039 eGenes (FDR < 1%, Supplementary Fig. 2)[20,21]. Stepwise regression analysis (after conditioning on the lead variant) identified a further 1702 independent exon-level islet *cis*-eQTLs in 1291 of the eGenes (21.3% of all eGenes), giving a total of 7741 eQTLs (Supplementary Data 1–4). At the 1291 eGenes with at least two independent exon–eSNPs, most primary eSNPs mapped closer to the canonical TSS than secondary eSNPs (Wilcoxon test $P = 6.3 \times 10^{-30}$): however, there were 503 (39.0%) of these genes for which the second eSNP identified during stepwise conditional analysis was more proximal to the TSS (Supplementary Fig. 2).

**Tissue-specific regulatory variation in islets**. For many complex traits of biomedical interest, the value of targeting the specific cell types of interest for dedicated eQTL discovery—as opposed to relying on existing eQTL data from more accessible tissues—remains unclear. To examine this, we considered the degree to which the set of 7741 exon-level islet eQTLs overlapped eQTLs detected in 44 tissues (N > 70) from version 6p of GTEx[8]. To allow direct comparison with InsPIRE, we reprocessed GTEx data to generate exon-level eQTLs ("Methods" section).

Of the 6039 islet eGenes, 5% (337) had no significant eQTLs, in either exon- or gene-level analyses in any of the 44 tissues examined in GTEx (Supplementary Data 5). We used *p*-value enrichment analysis ($\pi_1$)[22] to measure the proportion of islet eQTLs shared with other GTEx tissues, generating estimates ranging from 0.40 (hypothalamus) to 0.73 (adipose). By comparing the *p*-value distributions across tissues, this analysis can detect evidence of sharing which does not depend on arbitrary statistical thresholds. We detected the expected positive linear relationship between $\pi_1$ measures and sample sizes for the respective tissues in GTEx[8] (Fig. 1a). However, $\pi_1$ enrichment only reached 0.65 and 0.57 (respectively) for skeletal muscle (n = 361), and whole blood (n = 338), the tissues with the largest representation in this version of GTEx (see "Influence of sample size" in Supplementary Discussion). Whole pancreas, often naively used as a surrogate for the T2D-relevant islet component, represented an imperfect proxy for islet ($\pi_1 = 0.65$ with islets). This is not purely a consequence of low sample sizes: tissues with slightly larger sample sizes showed both higher ($\pi_1 = 0.71$, artery) and lower sharing of eQTLs ($\pi_1$ 0.64, skin no sun exposed). These data demonstrate that there is a component of tissue-specific genetic regulation that could, at these sample sizes, only be detected in islet, illustrating the value of extending current expression profiling efforts to additional tissues and cell types of particular biomedical importance.

**Cellular heterogeneity**. The human islets analyzed in this, and other, studies include a mixture of cell types, including the hormone-producing α, β and δ-cells, and a variable amount of adherent exocrine material. From the perspective of T2D pathogenesis, the transcriptomes of the endocrine compartment are of most interest. However, the eQTLs identified could have their origins from any of the cellular components. We used a number of approaches to address interpretative challenges resulting from this cellular heterogeneity.

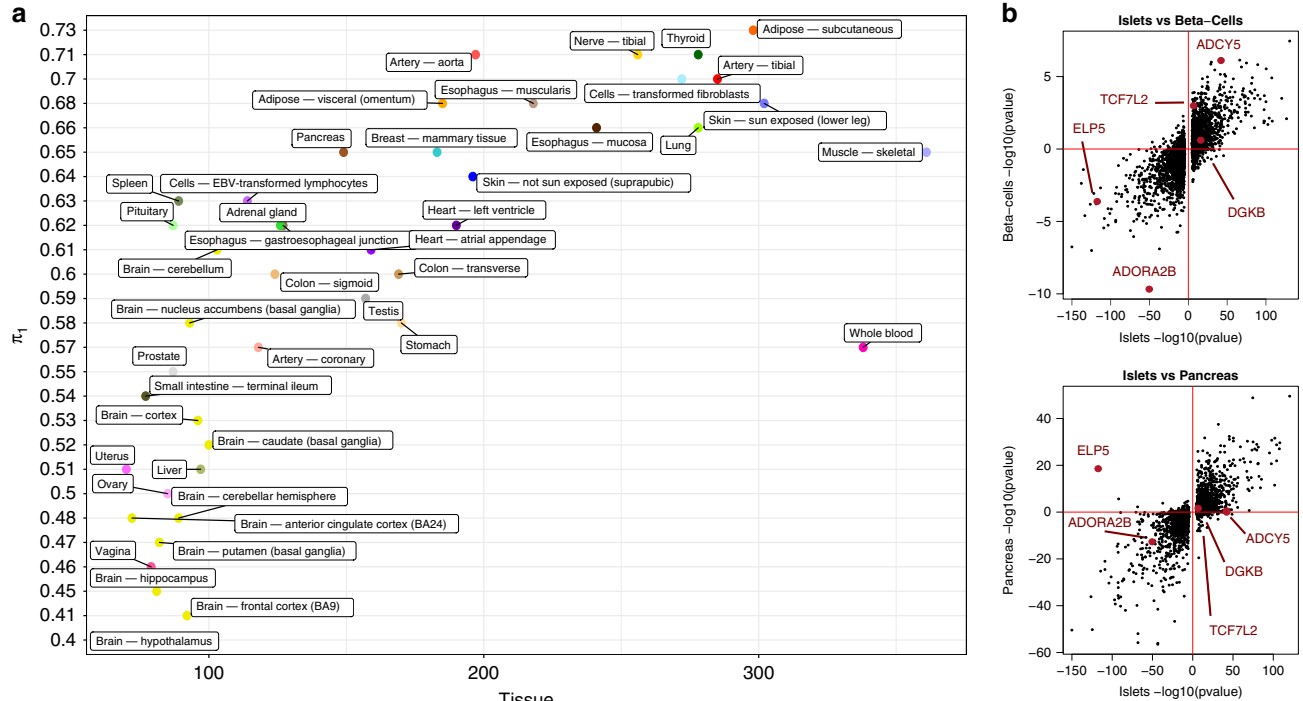

**Fig. 1 Islet eQTLs and their activity in other tissues. a** Proportion of islet eQTLs active in GTEx tissues using *p*-value enrichment analysis ($\pi_1$ estimate for replication). **b** Comparison between eQTLs discovered in islets and their *p*-values in β-cells (top figure, $N = 26$) and whole pancreas tissue from GTEX (bottom figure, $n = 149$). The axes show the $-\log_{10}$ *p*-value of the eQTL associations adjusted by the eQTL direction (positive or negative) of effect with respect to the reference allele. Source data are provided as a Source data file.

First, we performed tissue deconvolution analysis to estimate the proportion of exocrine contamination: these analyses were performed prior to the PC adjustment used to generate the main results and used reference expression signatures for exocrine pancreas, β-cells, and islet non-β-cells. The last two of these were generated from a subset ($n = 26$) of the islet preparations after FAC sorting using the zinc-binding dye Newport Green[18] ("Methods" section and in Supplementary Discussion). Estimates of the proportion of exocrine pancreas contamination ranged from 1.8 to 91.8% (median 33.5%): these were significantly correlated (spearman correlation, $\rho = -0.406$, $P = 1.4 \times 10^{-11}$) with independent estimates of exocrine content obtained at islet collection by dithizone staining ($n = 232$; Supplementary Fig. 3). Within the islet endocrine fraction, median estimates of β-cell (58.8%, IQR 43.2–66.9%) and non-β-cell (41.2%, 33.1–56.8%) fractions are in agreement with estimates from morphometric assessment[23]. In 37 samples from donors annotated as having T2D, median estimates of β-cell composition were lower than those from nondiabetic donors ($n = 330$; linear model, $P = 3.3 \times 10^{-2}$, Supplementary Fig. 3). This provides independent support, based on transcriptomic signatures, of evidence, from morphometric and physiological studies, that the functional mass of β-cells is reduced in T2D[24,25].

Of the 420 InsPIRE samples, β-cell-enriched transcriptomes were available for 26 following FAC sorting. With this limited sample size, the only eQTL reaching significance, and then only at a less stringent threshold of FDR < 5% (Supplementary Data 6) was at *ADORA2B* ($P = 3.8 \times 10^{-10}$, slope = −1.20): this signal was also detected in InsPIRE islets ($P = 3.9 \times 10^{-51}$, slope = −0.65) and GTEx pancreas ($P = 1.6 \times 10^{-16}$, slope = −0.73; Supplementary Fig. 4 and Supplementary Data 7). By comparing the *p*-value distributions of the eQTLs in islets vs β-cells[22], we estimate that 46% ($\pi_1 = 0.46$) of islet eQTLs are active in β-cells (Fig. 1b). By reevaluating significance for eQTLs in β-cell

association results using the 7741 independent significant SNP–exon pairs, we were able to replicate 227 islet eQTLs in β-cells (FDR < 1%, Supplementary Data 8). Genes with cell-type-specific regulatory effects were sought by testing for interactions between genotype and cellular fraction estimates, controlling for technical variables ("Methods" section). We identified 18 islet *cis*-eQTLs with a genotype-by-β-cell proportion interaction and eight with a genotype-by-exocrine cell proportion interaction (FDR < 1%, Supplementary Data 9, 10 and 11).

We conclude that a substantial proportion of the regulation of gene expression detected in pancreatic islets derives from cell-type-specific effects. Ongoing efforts to develop a single-cell view of islet transcriptional signatures should inform these analyses, although the limited sample size of current studies[26–30] and the paucity of genotype information means they offer little direct insight into the relationship between genetic variation and cell-type-specific transcript abundance.

**Functional properties of islet genetic regulatory signals**. Using previously published islet chromatin states derived from histone modification data[15], we observed a significant enrichment of islet eSNPs in active islet chromatin states, including active TSS (fold enrichment = 3.84, $P = 5.5 \times 10^{-206}$), active enhancers (fold enrichment > 1.73, $P < 4.8 \times 10^{-04}$ between active/inactive enhancer states), and stretch enhancers (fold enrichment = 1.57, $P = 2.7 \times 10^{-13}$), with concomitant depletion of eSNPs in repressed and quiescent states (fold enrichment < 0.66; Supplementary Fig. 5 and Supplementary Table 1). This recapitulates the enrichment observed for T2D GWAS signals within active islet chromatin (Supplementary Fig. 6 and Supplementary Table 2)[11,15,31,32]. Next, we examined the relationship between the chromatin context of islet eSNPs and their effect sizes (Fig. 2a): not only that the active TSS chromatin states showed high enrichment of eSNPs, but that

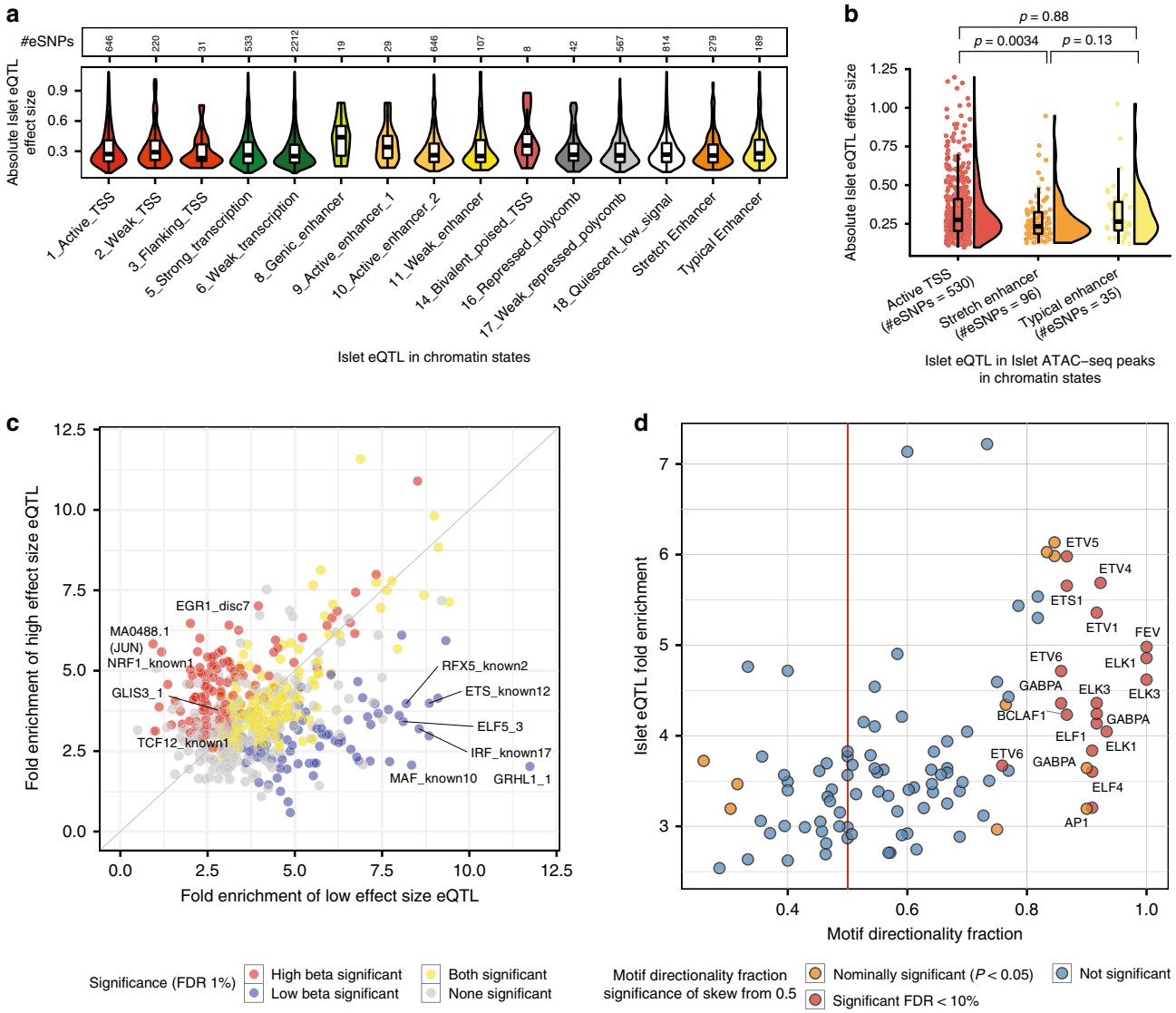

**Fig. 2 Integration of islet eQTL with epigenomic information. a** Distribution of absolute effect sizes for islet eQTLs in each islet chromatin state.
**b** Distribution of absolute effect sizes for islet eQTL in ATAC-seq peaks in three islet chromatin states. eQTL SNPs in ATAC-seq peaks in stretch enhancers
have significantly lower effect sizes than SNPs in ATAC-seq peaks in active TSS and typical enhancer states. *P*-values were obtained from a Wilcoxon rank-
sum test. **c** Fold enrichment for transcription factor footprint motifs to overlap low vs high effect size islet eQTL SNPs. **d** TF footprint motif directionality
fraction vs fold enrichment for the TF footprint motif to overlap islet eQTLs. TF footprint motif directionality fraction is calculated as the fraction of eQTL
SNPs overlapping a TF footprint motif, where the base preferred in the motif is associated with increased expression of the eQTL eGene. Significance of
skew of this fraction from a null expectation of 0.5 was calculated using the binomial test. Source data are provided as a Source data file.

the eSNPs that overlap active TSS chromatin states had larger
effects than those in repressed or weak-repressed polycomb
states (Wilcoxon rank-sum test $P = 3.9 \times 10^{-2}$).

A major portion (65.7%) of the islet-active TSS chromatin state
territory is occupied by islet ATAC-seq peaks. When we focused
the analysis on eSNPs within islet ATAC-seq peaks (Supplemen-
tary Fig. 7 and Supplementary Table 3), those within stretch
enhancers (islet-specific enhancer chromatin state segments >3 kb
(ref. [29])) had smaller effects than those in active TSSs (Wilcoxon
rank-sum test $P = 3.4 \times 10^{-3}$; Fig. 2b). Given the inverse
relationship between eQTL effect size and the number of samples
required to identify significant association, one corollary is that
eSNPs in tissue-specific stretch enhancers, which have smaller
effect sizes compared to those in more ubiquitous TSSs, are likely
to require larger sample sizes for eQTL discovery.

We previously reported enrichment of selected TF footprint
motifs at islet eSNPs[15]. Here, with a larger eSNP catalog, we sought
to determine how eSNP effect size and target gene expression
directionality is associated with base-specific TF-binding prefer-
ences. Using published TF footprint data (in vivo-predicted TF
motif binding) from human islet ATAC-seq analyses[15], we
partitioned eSNPs into two equally sized bins (absolute slope ≥
vs < 0.254 standard deviation units). Higher effect size eSNPs were
preferentially enriched (<1% FDR) for footprint motifs character-
istic of islet-relevant TF families, such as GLIS3 (motif GLIS3_1,
$P = 1.2 \times 10^{-6}$). Other footprint motifs, including the RFX and
ETS families of TFs, were significantly enriched for low effect size
eSNPs ($P < 2 \times 10^{-4}$; Fig. 2c and Supplementary Data 12).

Finally, since TFs can act as activators, repressors, or both[33],
we asked, using previously published massively parallel reporter

assay (MPRA) data from HepG2 and K562 cell lines[34], whether eSNP alleles matching the base preference at TF footprint motifs have a consistent directional impact on gene expression. We defined a motif directionality fraction score (ranging from repressive [0] to activating [1]) for each TF footprint motif ("Methods" section). Of the 109 motifs reported as consistently activating or repressive across HepG2 and K562 cell lines that were also present in our study, 16.5% ($n = 18$) showed skewed activator preference in islets (<10% FDR; Fig. 2d, Supplementary Fig. 8 and Supplementary Table 4). The activator motifs we identified include many ETS family members that have a known preference for transcriptional activation[34].

Our analyses demonstrate the value of contrasting tissue-specific stretch enhancers with more ubiquitous TSS states to delineate the role of underlying chromatin on function, and illustrate how the integration of eQTL information with ATAC-seq and high-resolution TF footprinting reveals the in vivo activities of these upstream regulators.

**Islet eQTLs are enriched among T2D and glycemic GWAS variants.** Diverse lines of evidence emphasize the contribution of reduced pancreatic islet function to the development of T2D, with many T2D GWAS loci acting primarily through reductions in insulin secretion[5,11,13,24]. To examine the relationships between tissue-specific regulation of gene expression and T2D predisposition alleles, we focused on 403 lead GWAS SNPs with the strongest associations to T2D in Europeans (as reported in Mahajan et al.[5]) and 56 variants significantly associated with T2D-relevant continuous glycemic traits, including fasting glucose and β-cell function (HOMA-B) in nondiabetic individuals (Supplementary Data 13)[3,35–37]. For comparison, we included 55 GWAS variants implicated in T1D predisposition[38]. To determine the extent to which the GWAS variants were selectively enriched for islet eQTL associations, we extracted exon-level eQTL information for each of these variants from InsPIRE and the 44 GTEx[8] tissues. We compared observed effect size estimates to those derived from a null distribution of 15,000 random eSNPs, matched to the GWAS SNPs with respect to the number of SNPs in linkage disequilibrium (LD), distance to TSS, number of nearby genes, and minor allele frequency (MAF; "Methods" section). Figure 3 shows the enrichment in eQTL effect sizes at T2D/glycemic GWAS-associated variants for five tissues implicated in T2D pathogenesis (subcutaneous adipose tissue, skeletal muscle, liver, islets, and plus hypothalamus), with pancreas and whole blood for comparison.

We detected nominally significant enrichment for islet eQTLs for variants associated with continuous glycemic traits (normalized enrichment score (NES) = 1.27; $P = 3.7 \times 10^{-3}$; Supplementary Fig. 9 and Supplementary Data 14): of all the tissues considered, islets generated the most significant enrichment for this phenotype. Islet cis-eQTLs were also enriched amongst the 403 T2D variants (NES = 1.09; $p = 5.3 \times 10^{-3}$), as were eQTLs from pancreas, skeletal muscle, adipose, and eight other GTEx tissues. In a subset of 43 variants (from the 403) considered to be mediated through defects in insulin secretion (thereby implicating islet dysfunction)[5,37], the degree of enrichment was increased (NES = 1.15, $p = 3 \times 10^{-2}$). No evidence of enrichment of islet eQTL signals was seen for T1D-risk variants (NES = 1.01; $p = 0.47$), consistent with the consensus that genetic risk for T1D is largely mediated through immune mechanisms[38]. These data reveal tissue-specific patterns of genetic regulatory impact for variants at T2D- and glycemic-trait loci similar to the mechanistic inferences generated by physiological analysis of those signals. They also highlight the importance of matching the tissue origin of the transcriptomic data used for mechanistic

inference to the tissue-specific impact of each GWAS signal on disease predisposition.

**Identifying effector transcripts for T2D and glycemic traits.** Previous studies have identified 27 GWAS signals displaying apparent overlap between islet eQTLs and the T2D/glycemic GWAS signals[15–17], but not all of these signals have been evaluated with respect to the statistical evidence for colocalization and not all coincident signals have replicated despite ostensibly similar designs and power[17]. Here, we took the opportunity offered by increases in the sample sizes available for both T2D GWAS[5] and islet cis-eQTL analyses, to undertake a systematic reanalysis to identify effector transcripts mediating the activity of T2D and glycemic traits GWAS variants. There are multiple methods for evaluating the evidence for colocalization, but these make different assumptions and often lead to discrepant results[39]. We focused on the colocalization evidence provided by two complementary algorithms: COLOC[40], which assesses differences in regression coefficients of variants associated to two traits, and RTC[41], which assesses the differences in ranking of SNPs associated with one trait after conditioning on the most associated SNP for the other.

We detected evidence for colocalization (using either method) for islet eQTLs at 46 GWAS loci (47 independent signals, Fig. 4b, Supplementary Figs. 10 and 11). Of those, signals at 22 loci were supported by both methods (constituting 23 signals, given 2 signals at *DGKB*): 21 of these signals reflect associations with T2D, and 6 with glycemic traits (4 were signals for both T2D and glycemic traits, Supplementary Data 15). Amongst this set of 23 signals, we confirmed colocalization with T2D or glycemic associations at several previously reported cis-eQTL signals, adding to the evidence for *ADCY5*, *HMG20A*, *CAMK1D*, and *DGKB*[11,15,17,42], as candidate effector transcripts. At 13 other signals, including *CLUAP1*, *EIF3C*, and *RNF6*, we observed islet cis-eQTL colocalization not reported before (Supplementary Data 15). At the remaining 24 (of 47) signals, colocalization was supported by either RTC or COLOC but not both, 19 of these reflecting associations with T2D, 8 with glycemic traits (3 overlapping). This included 7 previously reported islet cis-eQTL signals (including those at *NKX6-3*, *IGF2BP2*, or *KLHL42*), plus 17 signals never reported before (including those at *ADRA2A*, *PDE8B*, and *SLC12A8*) (Supplementary Table 5).

For example, previous efforts to characterize the mechanism of action at the *TCF7L2* locus have demonstrated that the fine-mapped T2D-risk allele at rs7903146 influences chromatin accessibility and enhancer activity in islets[43], but evidence linking these events to *TCF7L2* expression was not previously detected in genome-wide eQTL studies. Our data reveal that the C allele of rs7309146 increases islet expression of the last (3′) exon of *TCF7L2* (eQTL slope = 0.21, $P = 1.9 \times 10^{-7}$, Fig. 3b). A directionally consistent signal ($p < 0.05$) was seen for 15 more of the 18 exons expressed. The same eQTL signal was also detected in the smaller β-cell-specific analysis ($n = 26$; eQTL slope = 0.72; $p = 1.0 \times 10^{-3}$). The association between rs7903146 and *TCF7L2* expression was restricted to islets, consistent with evidence that nondiabetic carriers of the *TCF7L2* risk-allele display markedly reduced insulin secretion[44]. Recent studies have proposed *ACSL5* (ref. [45]) as a possible effector transcript at this locus, but we found no support—in any tissue—that rs7903146 influences *ACSL5* expression.

Of the 27 previously reported signals of GWAS/islet eQTL overlap (Supplementary Table 5), 12 were not observed in our exon-eQTL-based analysis. Potential explanations include low expression level of targeted genes, and between-study differences in analytical approach and significance thresholds. For example,

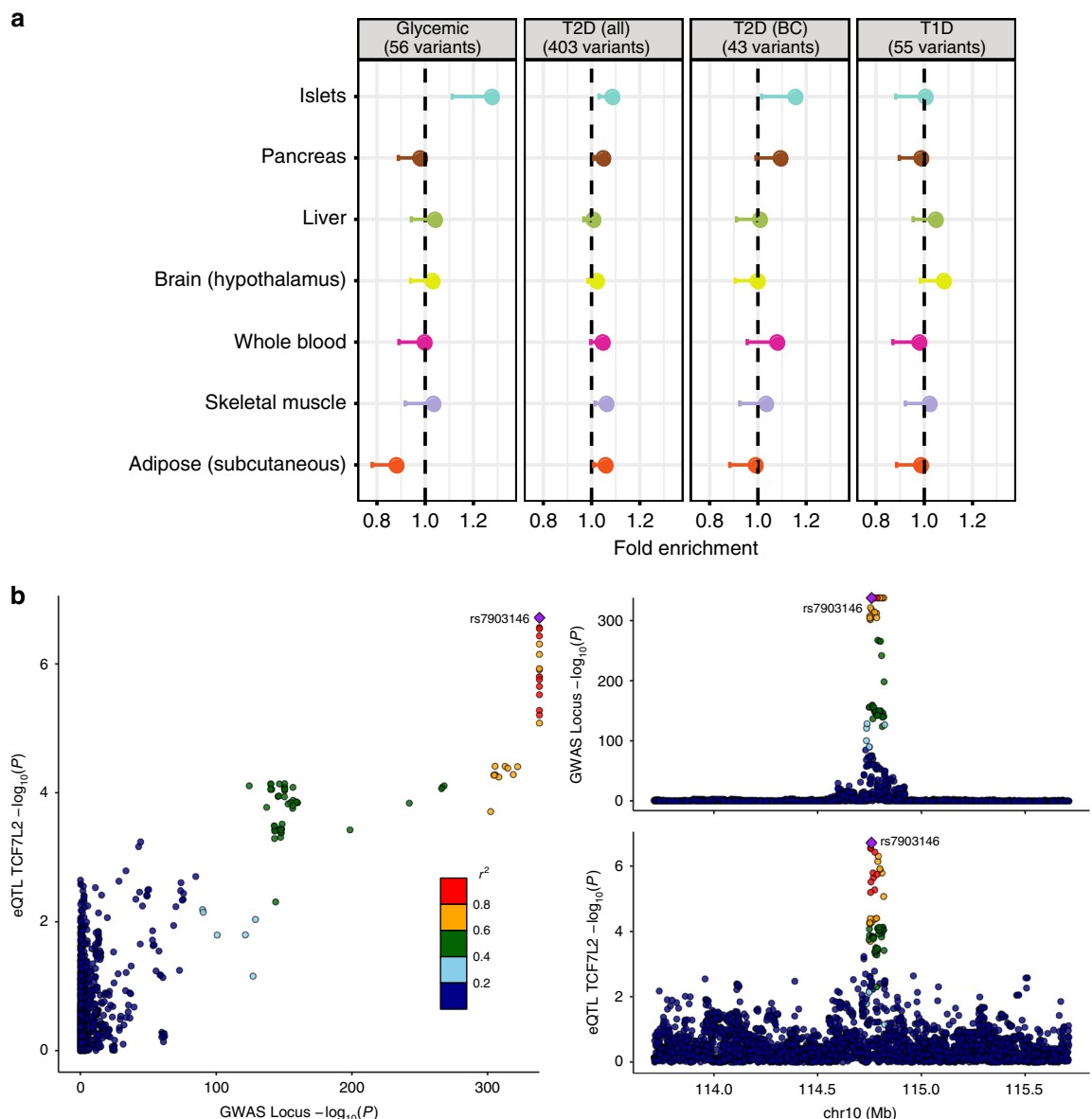

**Fig. 3 GWAS SNPs in islet eQTLs. a** Enrichment of eQTL effect sizes in different GTEx tissues at T2D (all) and glycemic GWAS-associated variants. Numbers within square brackets denote the number of variants implicated for each the trait. Also shown a subset of T2D GWAS associated with reduced insulin secretion or islet β-cell dysfunction (T2D (BC)) and type 1 diabetes (T1D)-associated signals. **b** LocusCompare plot for the T2D GWAS p-values in the *TCF7L2* locus. Plots on the right −log10 p-values for the GWAS (top) and for the the eQTL for *TCF7L2*, highlighting in both the GWAS lead SNP in the *cis* window tested for eQTLs. On the left it shows a comparison of the p-values in both analyses. Source data are provided as a Source data file.

the primary eQTL for *MTNR1B* has shown consistent islet *cis*-eQTL signals in previous studies[17,46], but the low level of expression of *MTNR1B* in islets meant that the exon-level read coverage fell below our threshold for inclusion. However, in the complementary gene-level analyses, there remained strong evidence of colocalization between the lead T2D variant (rs10830963) and *MTNR1B* expression ($p = 5.3 \times 10^{-21}$; Supplementary Data 2). At *ZMIZ1*, the previously reported *cis*-eQTL was nominally significant (rs185040218; $p = 3.0 \times 10^{-5}$), but did not reach the 1% FDR threshold for inclusion in colocalization testing.

At some loci, complex, but divergent, patterns of association between the eQTL and T2D GWAS signals challenged the assumptions of these colocalization methods. At the *MAP3K11* locus, for example, the association plots indicate two independent

islet eQTL signals for *LTBP3* (rs11227223 and rs1194077), but only the latter signal colocalizes with the T2D GWAS signal at rs1783541 (Supplementary Fig. 10). RTC detects this as colocalization as it controls for additional eQTL signals, but this was not possible with COLOC that assumed a single eQTL variant to be active.

We further attempted to characterize eGenes that overlapped T2D/glycemic GWAS signals by assessing the impact of changes in glycemic status on islet expression. We used data from a recent analysis of islets recovered from diabetic and nondiabetic donors, focussing on transcripts that showed acute changes in expression when exposed to glucose levels in culture that contrasted with those to which they had been habituated[47] (Supplementary Data 16). Islet eGenes, such as *STARD10*, *WARS*, *SIX3*, *NKX6-3*, and *KLHL42*, which may be of particular interest in that their

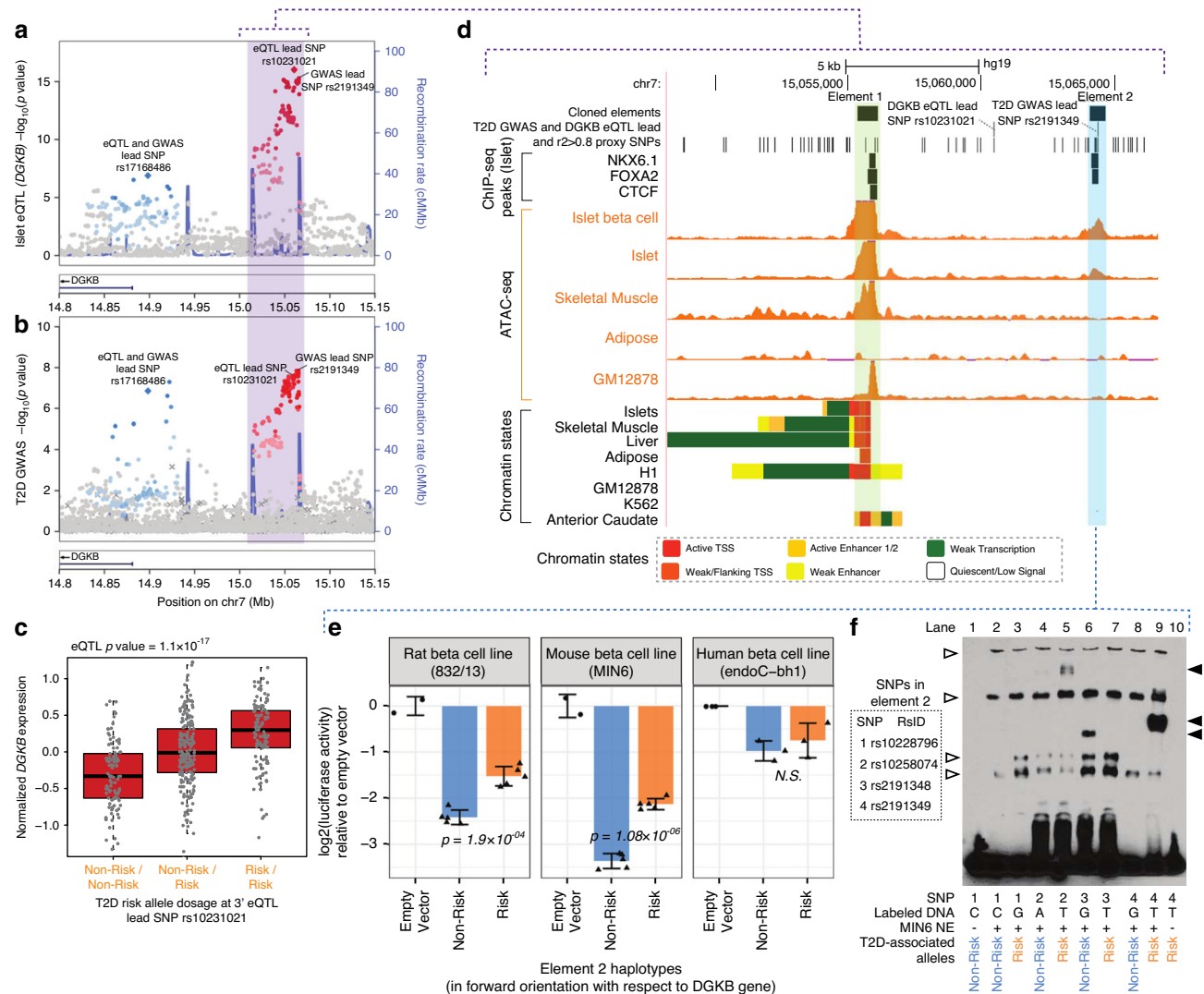

**Fig. 4 Functional assessment of *DGKB* eQTL locus. a** We show the two of the three independent islet eQTL signals that colocalize with identified independent GWAS variants near the *DGKB* gene locus (lead SNP rs17168486 referred at as the 5′ signal and lead SNP rs10231021 referred to as the 3′ signal). These signals colocalize with two independent T2D GWAS signals shown in **b**, where rs17168486 is referred to as the 5′ signal and lead SNP rs2191349 referred to as the 3′ signal. LD information was not available for SNPs denoted by (×). The third GWAS variant and the third eSNP are not shown as both are located outside this region and in opposite location with respect to DGKB, showing no evidence of colocalization. **c** Normalized *DGKB* gene expression levels relative to the T2D-risk-allele dosage at the 3′ islet eQTL for *DGKB* lead SNP rs10231021. eQTL *p*-value adjusted to the beta distribution is shown. **d** Genome browser view of the region highlighted in purple in **a** and **b** that contains the 3′ *DGKB* eQTL and T2D GWAS signals. Two regulatory elements (element 1 highlighted in green, element 2 highlighted in blue) overlapping ATAC-seq peaks in islet β-cells (islet single nuclei ATAC-seq[49]) and bulk islets (islet track represents one islet sample from Varshney et al.[15]) were cloned into a luciferase reporter assay construct for functional validation. All ATAC-seq tracks are normalized to 10 M reads and scaled from 0–15. **e** Log 2 luciferase assay activities (normalized to empty vector) are shown for in rat (832/13), mouse (MIN6), and human (EndoC–βH1) β-cell lines for the element 2 (cloned in the forward orientation), highlighted in blue in **d**. The risk haplotype shows significantly higher ($p < 0.05$) activity than the non-risk haplotype in 832/13 and MIN6, consistent with the eQTL direction shown in **c**. *P*-values were determined using unpaired two-sided *t*-tests. **f** EMSA for probes with risk and non-risk alleles at the four SNPs overlapping the regulatory element validated in **e**, using nuclear extract from MIN6 cells. Filled arrows, allele-specific binding; open arrows, non-allele-specific binding of proteins to probes. Source data are provided as a Source data file.

expression in islets is regulated both by T2D-associated variation and by acute changes in glucose exposure.

**Experimental validation at *DGKB*.** The *DGKB* locus features three independent T2D GWAS signals. In the T2D meta-analysis of Mahajan et al.[5], these are represented by the lead SNPs rs2191349, rs17168486, and rs2908334. In the current study, we also report three independent islet *cis*-eQTLs influencing *DGKB* expression represented by the lead eSNPs rs10278505, rs17168486, and rs10231021. There is colocalization of these associations at

two signals: we refer to these as the 5′ signal (rs17168486) and the 3′ signal (rs10231021, which is in perfect LD with rs2191349 in Europeans ($r^2 = 1$, $D' = 1$); Fig. 4a). There was no evidence for colocalization of the third GWAS signal (located on the 3′ end of the region (15.2 Mb)) and the third eSNP (located in the far 5′ of the region (14.3 Mb)): these are not discussed further. For both colocalizing GWAS signals, the T2D-risk allele is associated with increased islet expression of *DGKB* (Fig. 4a–c), and physiological analyses for these variants are consistent, with mediation through islet dysfunction[5,48].

For functional follow-up of the 3′ signal (Fig. 4a, b), we considered seven variants that mapped to ATAC-Seq peaks in bulk islets and islet β-cells (islet single nuclei ATAC-seq[49]) and were in high LD ($r^2 > 0.8$) with rs2191349 (Fig. 4d). Three (rs7798124, rs7798360, and rs7781710, Fig. 4d, element 1) overlap an ATAC-seq peak shared across islets/β-cells, skeletal muscle, and the lymphoblastoid cell line GM12878 (ref. [50]): four others (rs10228796, rs10258074, rs2191348, and rs2191349, Fig. 4d, element 2) lie in a smaller but more islet/β-cell-specific peak 8145 bp away. We cloned these putative regulatory elements into luciferase reporter constructs and performed transcriptional reporter assays in three widely used cellular models of the β-cell (human EndoC-βH1 (ref. [51]), rat INS-1-derived 823/13, and mouse MIN6 (ref. [52]); see "Methods" section). Element 1 showed consistently high enhancer activity across all three lines (when cloned in the forward orientation), but no allelic differences consistent with the eQTL direction of effect (Supplementary Fig. 12). Element 2 showed reduced luciferase expression in all three β-cell lines when in forward orientation with respect to DGKB (Fig. 4e). The T2D-risk haplotype showed higher expression than the non-risk haplotype in 832/13 ($p = 1.9 \times 10^{-4}$) and MIN6 ($p = 1.1 \times 10^{-6}$; Fig. 4e), which is consistent with the eQTL direction (Fig. 4c). Equivalent data for the human EndoC-βH1 cell line was directionally consistent but not significant (Fig. 4e). Luciferase assays using element 2 in reverse orientation also showed consistent trends across the cell lines, reaching significance in 832/13 (Supplementary Fig. 13). In electrophoretic mobility shift assays (EMSAs) performed using MIN6 nuclear extract, three element 2 variants (rs10228796, rs2191348, and rs2191349) showed allele-specific binding to nuclear proteins (Fig. 4f, filled arrows), supporting a functional regulatory role for all these variants. Given that the three variants are in a relatively small region and show allelic differences in binding to proteins, one explanation is that a complex of TFs binds to this regulatory element, with the T2D-risk alleles also alleviating the regulatory element repression in a direction consistent, with the observed effects of the 3′ signal on DGKB expression (Fig. 4c).

At the 5′ eQTL, we focused attention on rs17168486, which was both the lead SNP for islet cis-expression and T2D association and located in an islet ATAC-seq peak (Supplementary Fig. 14a). However, luciferase reporter constructs found no consistent allelic effects on transcriptional activity (Supplementary Fig. 14b).

## Discussion

We have used transcriptome sequencing in 420 human islet preparations to address issues of general relevance to the mechanistic interpretation of noncoding association signals detected by GWAS. We report an increase in the catalog of eQTLs from pancreatic islets from ~4000 published in Varshney et al.[15] to >7000 and document the degree to which RNA-Seq of a disease-relevant tissue missing from a reference set (e.g., GTEx) provides a more complete survey of islet eQTLs. We used this information to extend the number of association signals for T2D and related glycemic traits from 27 loci overlapping with islet eQTLs to 46 (47 signals, including 23 signals supported by two different methods), identifying candidate effector transcripts at several loci. We also explored how cellular heterogeneity (both within the tissue of interest, and reflecting contamination with cells not of direct relevance) can complicate the interpretation of GWAS signal colocalization. We integrated our eQTL catalog with islet epigenomic data to reveal effect size heterogeneity attributable to local chromatin context and to infer in vivo TF directional activities.

Analyses of the physiological association patterns and regulatory annotation enrichment signals of T2D-risk alleles indicate that many, though by no means all, act through the islet[9,10,12,14,53]. A major motivation behind development of this enhanced catalog of islet eQTLs was to support identification of effector transcripts mediating the downstream consequences of these noncoding alleles. At DGKB, for example, evidence that both the T2D signals colocalize with islet eQTLs with directionally consistent impacts on DGKB expression lends credibility to a causal role for DGKB in T2D predisposition.

However, it is important to emphasize that robust inference from the coincidence of eQTLs and GWAS signals is difficult. First, the expression data in our study was derived from human islets cultured in basal glycemic conditions: eQTL signals restricted to a subset of the cells within those islets would have been hard to detect, and the same is true for genes whose expression is dependent on stimulation. Since not all T2D loci act through the mature islet, some of the eQTLs detected may reflect tissue-specific regulation that is not germane to the development of the diabetic phenotype.

Second, confident assignment of colocalization can be difficult. There are multiple algorithms to assess the evidence that two association signals are likely to reflect the same causal variants, but agreement between them is incomplete[39]. An additional challenge arises from the complex architecture of many GWAS signals, such that conditional decomposition is required before colocalization across multiple overlapping signals can be accurately assigned[54]. This is especially important when the sets of GWAS and cis-eQTL signals at a given locus are not completely overlapping, since obvious colocalization at one of the contributing signals can be masked by differences in the overall shape of the association signals that confounds simplistic analysis.

Third, recent studies have shown that functionally constrained genes—which are depleted for missense or loss-of-function variants—are also less likely to have eQTLs, indicating uniform intolerance of both regulatory and coding variation[55–57]. Complementary studies focusing on regulatory elements have shown that large, cell-specific stretch enhancers harbor smaller effect size eQTLs than ubiquitous promoter regions[58] and that genes with more cognate enhancer sequence are depleted for eQTLs[57]. Our finding that islet eQTLs that map to the islet stretch enhancers most frequently implicated in GWAS regions had smaller eQTL effect sizes is consistent with these observations.

Finally, it is critical to emphasize that, even when colocalization has been demonstrated between a GWAS variant and a tissue-appropriate eQTL signal, this does not constitute proof that the eGene concerned mediates disease predisposition. Causal relationships other than variant-to-gene-to disease are possible, including the possibility the variant has horizontally pleiotropic effects on each[59]. Growing understanding of the extent of shared local regulatory activity and regulatory pleiotropy makes such an alternative explanation all the more credible[60]. It is best to regard genes highlighted by coincident GWAS and eQTL signals as candidate effector transcripts, and to proceed to experimental approaches that enable direct tests of causality. These may involve perturbing the gene across a range of disease-relevant cell lines and animal models, and determining the impact on phenotypic readouts that represent reliable surrogates of disease pathophysiology.

## Methods

**Cohort characteristics**. The samples from 420 donors included 189 males and 231 females, with an age range of 16–81 years (median = 54 years, 11 not available). Of the 420 individuals, 37 were identified as diabetic. BMI information was available for 334 individuals (median, 26.3 kg/m$^2$), while HbA1c measurements were available for 198 (median, 5.8%). Due to the historical nature of some of the

samples used in this study, QC information about the pancreatic islet isolation was limited: 254 samples listed their purity (median, 75%) and 26 samples listed their viability (median, 93.5%). No other biological information was available in the historical records. This information is available in the covariate files included in the EGA submission.

**Pancreatic islet sample collection and processing**. Samples collection and processing is summarized in Supplementary Fig. 17.

The processing of the Geneva samples from Nica et al.[18] were originally done using RNA libraries with 49-bp paired-end reads; however, in order to allow joint analysis with the other available datasets for this study, mRNA samples were reprocessed using a 100-bp paired-end sequencing protocol. The library preparation and sequencing followed customary Illumina TruSeq protocols for next generation sequencing as described in the original publication[18]. All procedures followed ethical guidelines at the University Hospital in Geneva.

The 89 Lund samples from Fadista et al.[16] were jointly processed with 102 new islet samples that were processed uniformly following the same protocol. These islet samples were obtained from 191 cadaver donors of European ancestry from the Nordic Islet Transplantation Programme (http://www.nordicislets.org). Purity of islets was assessed by dithizone staining, while measurement of DNA content and an estimate of the contribution of exocrine and endocrine tissue were assessed as previously described[61]. Total RNA was isolated with the AllPrep DNA/RNA Mini Kit following the manufacturer's instructions (Qiagen), sample preparation was performed using Illumina's TruSeq RNA Sample Preparation Kit according to manufacturer's recommendations. The target insert size of 300 bp was sequenced using a paired-end 101 bp protocol on the HiSeq2000 platform (Illumina). Illumina Casava v.1.8.2 software was used for base calling. All procedures were approved by the ethics committee at Lund University.

The Oxford dataset included samples collected in Oxford and Edmonton that were jointly sequenced in Oxford are included in this set of samples. Islet sample procurement, mRNA processing, and sequencing procedure has been described in van de Bunt et al.[17]. To the 117 samples previously published (78 from Edmonton and 39 from Oxford), 57 samples were added and processed following similar protocols as before (27 from Edmonton and 30 from Oxford). Briefly, freshly isolated human islets were collected at the Oxford Centre for Islet Transplantation in Oxford, or the Alberta Diabetes Institute IsletCore (www.isletcore.ca) in Edmonton, Canada. Additional islets were obtained from the Alberta Diabetes Institute IsletCore's long-term cryopreserved biobank. Freshly isolated islets were processed for RNA and DNA extraction after 1–3 days in culture in CMRL media. Cryopreserved samples were thawed as described in Manning et al.[62] and Lyon et al.[63]. RNA was extracted from human islets using Trizol (Ambion, UK or Sigma-Aldrich, Canada). To clean remaining media from the islets, samples were washed three times with phosphate-buffered saline (Sigma-Aldrich, UK). After the final cleaning step 1 mL Trizol was added to the cells. The cells were lysed by pipetting immediately to ensure rapid inhibition of RNase activity and incubated at room temperature for 10 min. Lysates were then transferred to clean 1.5 mL RNase-free centrifuge tubes (Applied Biosystems, UK). RNA quality (RIN score) was determined using an Agilent 2100 Bioanalyser (Agilent, UK), with a RIN score > 6 deemed acceptable for inclusion in the study. Samples were stored at −80 °C prior to sequencing. PolyA selected libraries were prepared from total RNA at the Oxford Genomics Centre using NEBNext ultra directional RNA library prep kit for Illumina with custom 8 bp indexes[64]. Libraries were multiplexed (three samples per lane), clustered using TruSeq PE Cluster Kit v3, and paired-end sequenced (100 nt) using Illumina TruSeq v3 chemistry on the Illumina HiSeq2000 platform. All procedures were approved by the Human Research Ethics Board at the University of Alberta (Pro00013094), the University of Oxford's Oxford Tropical Research Ethics Committee (OxTREC Ref. 2−15), or the Oxfordshire Regional Ethics Committee B (REC reference: 09/H0605/2). All organ donors provided informed consent for use of pancreatic tissue in research.

The USA samples from Varshney et al.[15] were originally processed as follows: 39 islet samples from organ donors were received from the Integrated Islet Distribution Program, the National Disease Research Interchange, and Prodo-Labs. Total RNA from 2000–3000 islet equivalents was extracted and purified using Trizol (Life Technologies). RNA quality was confirmed with Bioanalyzer 2100 (Agilent); samples with RNA integrity number (RIN) > 6.5 were prepared for mRNA sequencing. We added External RNA Control Consortium spike-in controls (Life Technologies) to one microgram of total RNA. PolyA+, stranded mRNA RNA-Seq libraries were generated for each islet using the TruSeq-stranded mRNA kit according to manufacturer's protocol (Illumina). Each islet RNA-Seq library was barcoded, pooled into 12-sample batches, and sequenced over multiple lanes of HiSeq2000 to obtain an average depth of 100 million 2 × 101 bp sequences. All procedures followed ethical guidelines at the National Institutes of Health (NIH).

**B-cell sample collection and processing**. To the 11 FAC-sorted beta-cells population samples previously published[18], we added 15 more samples that were processed following the same protocols. Briefly, islets were dispersed into single cells, stained with Newport Green, and sorted into "beta" and "non-beta" populations. The proportion of beta (insulin), alpha (glucagon), and delta (somatostatin) cells in each population (as percentage of total cells) was determined by

immunofluorescence. mRNA extractions as well as sequencing followed the same details described for islets samples processing for the Geneva samples.

**Read mapping and exon quantification**. The 100-bp sequenced paired-end reads were mapped to the GRCh37 reference genome[65] with GEM[66]. Exon quantifications were calculated for all elements annotated in GENCODE[67] v19, removing genes with >20% zero read count. All overlapping exons of a gene were merged into meta-exons with identifier of type ENSG000001.1_exon.start.pos_exon.end.pos, as described in Lappalainen[21]. Read counts over these elements were calculated without using read pair information, except for excluding reads where the pairs mapped to two different genes. We counted a read in an exon if either its start or end coordinates overlapped an exon. For split reads, we counted the exon overlap of each split fragment, and added counts per read as 1/(number of overlapping exons per gene). Gene-level quantifications used the sum of all reads mapped to exons from the gene. Genes with >20% zero read counts were removed.

**Genotype imputation**. Genotypes for all islet samples, including 19 β-cell samples, were available from OmniExpress and Omni2.5 genotype arrays. Quality of genotyping from the shared SNPs in both arrays was assessed before imputation separately and, samples were excluded if they had an overlap genotype success rate <90%; and MAF differences >20% compared to the 1000 G reported European MAF. The two panels were separately pre-phased with SHAPEIT[68] v2 using the IMPUTE2-supplied genetic maps. After pre-phasing, the panels were imputed with IMPUTE2 (ref. [69]) v2.3.1 using the 1000 Genomes Phase I integrated variant set (March 2012) as the reference panel[70]. SNPs with INFO score > 0.4 and HWE $p$ > 1e−6 (for chrX this was calculated from female individuals only) from each panel were kept. A combined vcf for each chromosome was generated from the intersection of the checked variants in each panel. Directly genotyped SNPs with a MAF < 1% (including the exome components of the chips not shared between all centers) were merged into the combined vcfs: (i) If SNPs were not imputed they were added and (ii) If SNPs had been imputed, the imputed calls for the individual were replaced by the typed genotype. Dosages were calculated from the imputation probabilities (genotyped samples) or genotype calls (WGS samples). For the 22 autosomes, the dosage calculation was: 2 × ((0.5*heterozygous call) + homozygous alt call). For chromosome X (where every individual should be functionally hemizygous), the calculation was: (0.5*heterozygous call) + homozygous alt call). Genotype calls for males can only be "0/0" and "1/1". The total number of variants available for analysis after quality assessment was 8,056,952.

For the 26 β-cell samples, 19 had genotypes available from OmniExpress arrays, whereas 7 had DNA sequence available. Variant calling from DNA sequence has been previously described in Nica et al.[18]. Briefly, the Genome Analysis Toolkit[71] v1.5.31 was used following the best practice variant detection v3 to call variants. Reads were aligned to the hg19 reference genome with BWA[72]. A confidence score threshold of 30 for variant detection was used and a minimum base quality of 17 for base calling. Good confidence (1% FDR) SNP calls were then imputed on the 1000 Genomes reference panel and phased with BEAGLE[73] v3.3.2. Imputation of variants from samples with arrays genotyping were imputed together with genotypes from individuals with islets samples as described before and then merged with genotypes from DNA sequences. SNPs with INFO score > 0.4, HWE $p$ > 1e−6 and MAF > 5%, were kept for further analysis. The total number of variants available for analysis after quality assessment was 6,847,993.

**RNA-Seq quality assessment and data normalization**. Heterozygous sites per sample were matched with genotype information to confirm the ID of the samples[74]. Eleven samples did not match with their genotypes, six of which would be resolved by identifying concordant matches. For the remaining samples, no matches were found on the genotypes and they were removed from the dataset, giving a total of 420 samples with genotypes. Raw read counts from exons and genes were scaled to ten million to allow comparison between samples with different libraries. Scaled raw counts were then quantile normalized.

We used PC analysis to evaluate and control the effects of unwanted technical variation, and the expected batch effects due to the islet sample processing and mRNA sequencing being performed across four labs. The main differences in samples processing, and sequencing differences were grouped in a variable identifying the lab of origin of the samples. Since each institution handled samples in a different way with different processing and sequencing protocols, we expect differences across samples from different labs to be greater than the differences between samples from the same lab. Supplementary Fig. 1 shows these differences, while comparing the samples distribution in PC1 vs PC2, with the colors identifyinand g the lab of origin (GEN, OXF, LUN, USA). This analysis also identified internal batch effects for OXF and LUN samples, as a second set of samples were sequenced for this study in both institutions. Therefore, to control for these and other potential sources of unwanted global variation, we included 25 PCs of expression, as covariates in the linear model used to identify eQTLs (see below).

To identify the optimal number of PCs require to control for differences in the origin of the samples for the discovery of eQTLs, we performed a permutation test: expression sample labels and expression covariates were permuted within each of the four laboratories before performing a standard eQTL analysis against non-permuted genotypes (and matched PCs for genotypes), using different numbers of

PCs for expression. Significant eQTLs beyond a 5% FDR in any of these analyses are considered false positives due to technical differences across laboratories. Our results indicate that controlling for ten PCs was sufficient to minimize the number of false positives due to batch effects originating from differences in processing of the islet samples. In addition, we performed an eQTL discovery controlling for 1, 5, 10, 20, 30, 40, and 50 PCs for expression, as well as gender, 4 PCs derived from genotype data, and a variable defining the laboratory of origin (coded as: OXF, LUN, GEN, and USA). After evaluation of the results, we conclude that controlling for 25 PCs for expression was optimal as this controlled for differences across and within laboratories, while maximizing the discovery of eQTLs.

**eQTL analysis**. eQTL analysis for islets and β-cells were performed using fastQTL[19] on 420 islet samples and 26 FAC-sorted β-cell samples with available genotypes. *Cis*-eQTL analysis was restricted to SNPs in a 1 MB window upstream and downstream the TSS for each gene, and SNPs with MAF > 1%. For the analysis of β-cell samples, we used a filter of MAF > 5%. Exon-level eQTLs identified best exon–SNP association per gene (using the –group flag), while gene-level eQTLs used gene quantifications and identified the best gene–SNP association. Variables included in the linear models were the first 4 PCs for genotypes, the first 25 PCs for expression, gender, and a variable identifying the laboratory of origin of the samples. Significance for the SNP–gene association was assessed using 1000 permutations per gene, correcting *p*-values with a beta approximation distribution[19]. Genome-wide multiple testing correction was performed using the *q*-value correction[22] implemented in largeQvalue[75].

Results of this joint analysis were highly correlated with those obtained from a fixed-effects meta-analysis of the four component studies, indicating appropriate control for the technical differences between the studies (Supplementary Fig. 16).

To discover multiple independent eQTLs, we applied a stepwise regression procedure has also been described in Brown et al.[76]. We tested significant eGenes (FDR < 1%) for secondary associations in any exon. The maximum beta-adjusted *p*-value (correcting for multiple testing across the SNPs and exons) over these genes was taken as the gene-level threshold. The next stage proceeded iteratively for each gene and threshold. A *cis*-scan of the window was performed in each iteration, using 1000 permutations and correcting for all previously discovered eQTLs. If the beta-adjusted *p*-value for the most significant exon–SNP or gene–SNP (best association) was not significant at the gene-level threshold, the forward stage was complete and the procedure moved on to the backward step. If this *p*-value was significant, the best association was added to the list of discovered eQTLs as an independent signal and the forward step proceeded to the next iteration. The exon-level *cis*-eQTL scan identified eQTLs from different exons, but reported only the best exon–SNP in each iteration. Once the forward stage was complete for a given gene, a list of associated SNPs was produced that we refer to as forward signals. The backward stage consisted of testing each forward signal separately, controlling for all other discovered signals. To do this, for each forward signal we ran a *cis*-scan over all variants in the window using fastQTL, fitting all other discovered signals as covariates. If no SNP was significant at the gene-level threshold the signal being tested was dropped, otherwise the best association from the scan was chosen as the variant that represented the signal best in the full model.

**GTEx eQTLs**. We identified exon-level eQTLs for 44 GTEx tissues using fastQTL[19] following the same procedure as for the islet eQTLs. Covariates included followed the previously published number of PCs for expression[8] and included 15 PCs for expression for tissues with <154 samples; 30 PCs for samples between 155 and 254 samples; and 35 PCs for samples with >254 samples. Independent eQTLs from exons were identified as described for islet eQTLs. The proportion of shared eQTLs between islet and β-cell eQTLs and the eQTLs from GTEx tissues were identified using $\Pi_1$[22].

**Tissue deconvolution**. To identify the contribution of β-cells, non-β-cells and exocrine (non-islet) cells to overall gene expression measured in islets, we performed an expression deconvolution analysis. Expression profiles from GTEx whole pancreas was used as a model for the exocrine component of expression[8], while FAC-sorted expression profiles from β-cell and non-β-cells from Nica et al.[18] were used to identify the fraction of expression derived from islet cells. First, we performed differential expression analysis of (a) exocrine vs whole islet samples; (b) β-cell vs whole islet samples; and (c) non-β-cell vs whole islet samples. The top 500 genes from each analysis were combined, and a deconvolution matrix of $\log_2$-transformed median expression values was prepared for each cell type. Next, deconvolution was performed using the Bioconductor package DeconRNASeq[77]. Deconvolution values per sample are included in the covariates file, together with the expression values in the EGA submission.

**Genotype-by-cell type associations**. Genotype-by-cell-type-specific regulatory effects were identified by testing for interactions between SNPs and cellular fraction estimates. We performed the analysis using a linear model and residuals from expression in gene quantifications. Residuals were extracted using a linear mixed model controlling for fixed effect variables (batch effects, islets purity, GC mean, and merge), and random variables (tag/primers and date of sequencing). The merge variable identified samples that were sequenced multiple times, with a final

set of reads merged from multiple files. Significance for the genotype-by-cell interactions were evaluated using FDR and 100 permutations in each analysis: genotype-by-β-cell proportions, genotype-by-non-β-cell proportions, and genotype-by-exocrine cell proportions.

**Enrichment of eQTLs in T2D and glycemic GWAS**. Within each tissue, we asked if the magnitude of the eQTL effect for a given set of GWAS SNPs were larger than expected for a randomly selected matched set of SNPs (as described below). We performed enrichment analysis for each trait and tissue type using the following procedure. For each trait, we used lead GWAS variants from the following sources: T2D (all; $n = 403$)[5]; subsets of these T2D-associated variants that likely act via β-cell action ($n = 43$); glycemic traits (fasting glucose and β-cell function (HOMA-B) in nondiabetic individuals; $n = 56$); and for comparison T1D ($n = 55$)[38] (Supplementary Data 13)[3,36,62]. As InsPIRE pancreatic islet and GTEx tissue-based estimates of the variant exon effect, we used eQTL beta coefficients for exons tested for each gene within 1 Mb of the GWAS lead variant. For each GWAS lead variant for a given trait, we identified the eQTL with the largest absolute effect size estimate among all the tested exons (max individual SNP beta). Across the lead GWAS variants, we took the median of the max individual SNP betas (observed median of the maxes). To generate a null distribution of the medians of the maxes, we repeated the analysis 15,000 times. For each replicate, we matched each GWAS SNP to a SNP present in the eQTL data based on the number of SNPs in LD, distance to TSS, number of nearby genes and MAF, and calculated the median of the maxes (null median of maxes). To form a distribution of the effect size enrichment we divided the observed median of the maxes by the null medians of the maxes. For each tissue and GWAS trait, we defined the median of the effect size enrichment distribution as the enrichment. We estimated the one-sided 95% confidence intervals as the fifth percentiles of the effect size enrichment distribution. We calculated the one-sided *p*-value for enrichment as the proportion of replicates with enrichment values <1.

**Colocalization of islet eQTL with T2D GWAS**. Colocalization of GWAS variants and eQTLs was performed using both COLOC[40] and RTC[19]. For the analysis using COLOC, all variants within 250 kb flanking regions around index variants were tested for colocalization, using default parameters from the software. The analysis used summary statistics from T2D GWAS[6] and fasting glucose[35]. GWAS variants and eSNPs pairs were considered to colocalize if the COLOC score for shared signal was >0.9. RTC analysis was also performed using defaults parameters from the software with a list of 459 lead GWAS variants for T2D and fasting glucose (Supplementary Data 13). Associations between GWAS and gene expression were considered as colocalizing if RTC score was >0.9. The plots showing colocalization of GWAS and eQTLs were generated using LocusCompare[54].

**Chromatin states, islet ATAC-seq, and TF footprints**. We used a previously published 13 chromatin state model that included pancreatic islets along with 30 other diverse tissues[15]. Briefly, these chromatin states were generated from cell/tissue ChIP-seq data for H3K27ac, H3K27me3, H3K36me3, H3K4me1, and H3K4me3, and input control from a diverse set of publically available data[31,78–80] using the ChromHMM program[81]. Chromatin states were learned jointly from 33 cell/tissues that passed QC by applying the ChromHMM (version 1.10) hidden Markov model algorithm at 200-bp resolution to five chromatin marks and input[15]. We ran ChromHMM with a range of possible states and selected a 13-state model, because it most accurately captured information from higher-state models and provided sufficient resolution to identify biologically meaningful patterns in a reproducible way. As reported previously[15], stretch enhancers were defined as contiguous enhancer chromatin state (active enhancers 1 and 2, genic enhancer, and weak enhancer) segments longer than 3 kb, whereas typical enhancers were enhancer state segments smaller than the median length of 800 bp (ref. [31]).

We downloaded raw ATAC-seq data (fastq files) for a total of 33 islet samples from nondiabetic donors: 14 from ref. [82], 17 from ref. [11], and two from ref. [15]. We processed these data uniformly by trimming all reads to 36 bp, mapping to hg19 using bwa-mem (version 0.7.15-r1140)[83], removing duplicates using Picard (http://broadinstitute.github.io/picard), and pruning reads to retain properly paired and mapped reads (samtools view -F 256 -F 1024 -F 2048 -q 30). Since these 33 samples were obtained from different studies, we sought to obtain the set of peaks that were reproducible. We subsampled each ATAC-seq sample bam file to the minimum read depth across samples of 27,994,993 reads, merged these subsampled bam files across all samples and called peaks, using (1) the merged bam file that uniformly represents each sample and (2) each individual sample bam file. We used MACS2 (https://github.com/taoliu/MACS), version 2.1.0, with flags -g hs–nomodel–shift -100–extsize 200 -B–broad–keep-dup all, to call peaks and filtered out regions blacklisted by the ENCODE consortium due to poor mappability (wgEncodeDacMapabilityConsensusExcludable.bed and wgEncodeDukeMapabilityRegionsExcludable.bed). We then selected peaks from the merged bam file that were reproducibly called across the majority (at least 17) of the 33 individual samples, resulting in 64,129 peaks that we used for downstream analyses.

TF footprint motifs are occurrences of TF motifs (obtained from databases of DNA-binding motifs for several TFs) in accessible chromatin regions (identified

from assays such as ATAC-seq). We downloaded previously published islet TF footprint motifs[15], which were generated in a haplotype-aware manner using ATAC-seq and genotyping data from the phased, imputed genotypes for two islet samples using vcf2diploid81 v0.2.6a and DNA-binding motif information for 1995 publicly available TF motifs[84–86]. We subset the footprint motifs selecting occurrences within the new set of reproducible ATAC-seq peaks described above.

**Filtering eQTL SNPs.** Since low MAF SNPs, due to low power, can only be identified as significant eQTL SNP (eSNPs) with high eQTL effect sizes (slope or the beta from the linear regression), we observed that absolute effect size varies inversely with MAF (Supplementary Fig. 15). To conduct eQTL effect size based analyses in an unbiased manner, we selected significant (FDR 1%) eSNPs with MAF >= 0.2. We then pruned this list to retain the most significant SNPs with pairwise LD ($r^2$) < 0.8 for the EUR population using PLINK[87] and 1000 genomes variant call format (vcf) files (downloaded from ftp://ftp.1000genomes.ebi.ac.uk/vol1/ftp/release/20130502/) for reference (European population). This filtering process resulted in $n = 3832$ islet eSNPs.

**Enrichment of genetic variants in genomic features.** To calculate the enrichment of islet eSNPs to overlap with genomic features, such as chromatin states, ATAC-seq peaks, and TF footprint motifs, we used the GREGOR tool[88]. For each input SNP, GREGOR selects ~500 control SNPs matched for MAF, distance to the gene, and number of SNPs in LD ($r^2$) ≥ 0.99. A unique overlap is reported if the feature overlaps any input lead SNP or its LD ($r^2$) > 0.99 SNPs. Fold enrichment is calculated as the number unique overlaps over the mean number of loci at which the matched control SNPs (or their LD ($r^2$) > 0.99 SNPs) overlap the same feature. This process accounts for the length of the features, as longer features will have more overlap by chance with control SNP sets. We used the following parameters in GREGOR for eQTL enrichment: $r^2$ threshold (for inclusion of SNPs in LD with the lead eSNP) = 0.99, LD window size = 1 Mb, and minimum neighbor number = 500. Since eQTL loci can only occur within ~1 Mb from TSSs of genes actually tested for eQTLs, we checked what fraction of the GREGOR control loci occurred within 1 Mb of tested genes. Out of total 6,031,279 control loci, 98.6% (5,949,654) variants occur within 1 Mb of the TSS for genes for which exon-level eQTL were tested.

For enrichment of T2D GWAS SNPs in islet chromatin states, we downloaded the list of T2D GWAS SNPs from Mahajan, et al.[5]. We pruned this list to retain the most significant SNPs with pairwise LD ($r^2$) < 0.2 for the EUR population using PLINK[87] and 1000 genomes variant call format (vcf) files (downloaded from ftp://ftp.1000genomes.ebi.ac.uk/vol1/ftp/release/20130502/) for reference (European population). This filtering process resulted in $N = 378$ T2D GWAS SNPs. We used GREGOR to calculate enrichment using the following specific parameters: $r^2$ threshold (for inclusion of SNPs in LD with the lead eSNP) = 0.8, LD window size = 1 Mb, and minimum neighbor number = 500.

We investigated if footprint motifs were more enriched to overlap eQTLs of high vs low effect sizes. We sorted the filtered (as described above) eQTL list by absolute effect size values and partitioned these into two equally sized bins ($N$ eSNPs = 1916). Since TF footprints were available for a large number of motifs ($N$ motifs = 1995), the enrichment analysis had a large multiple testing burden and limited power with 1916 eSNPs in each bin. Therefore, we only considered footprint motifs that were significantly enriched (FDR < 1%, Benjamini and Yekutieli method from R $p$. adjust function, $N$ motifs = 283) and that overlap the bulk set of eSNPs (LD $r^2$ < 0.8 pruned but not MAF filtered, $N$ eSNPs = 6,468, Supplementary Data 12) for enrichment with the binned set of eSNPs. This helped reduce the multiple testing burden. We then calculated enrichment for the selected footprints to overlap SNPs in each bin using GREGOR with same parameters as described above (Supplementary Data 12).

**eSNP effect size distribution in chromatin states.** We identified the islet eQTL eSNPs (after LD pruning and MAF filtering as described above) occurring in chromatin states or ATAC-seq peaks within chromatin states using BEDtools intersect[89]. Similar to the enrichment calculation procedure, we considered a unique eQTL overlap if the lead eSNP or a proxy SNP with LD ($r^2$) > 0.99 occurred in these regions. We considered the effect size as the slope or the beta from the linear regression for the eQTL overlapping each region. $P$-values were calculated using the Wilcoxon rank-sum test in R (ref. [90]).

**TF motif directionality analysis.** We selected TF footprint motifs that were significantly enriched (after Bonferroni correction accounting for 1995 total motifs) to overlap islet eQTL (considering LD $r^2$ < 0.8 pruned lead eSNPs or their $r^2$ > 0.99 proxy eSNPs as a locus) with at least ten eQTL locus overlaps, resulting in $N = 329$ TF motifs. We determined the overlap position of the eSNP with each TF footprint motif. We considered instances where the eSNP overlapped the TF footprint motif at a position with information content > =0.7 and either the eSNP effect or the non-effect allele was the most preferred base in the motif. For each TF footprint motif and eSNP overlap, we rekeyed the direction of effect on the target gene being positive or negative, with respect to the most preferred base in the motif. For each TF motif, we compiled the fraction of instances where the SNP allele that was most preferred in the TF footprint motif (i.e., base with highest probability in the motif) associated with

increased expression of the associated gene. We refer to this metric as the motif directionality fraction where fractions near 1 suggest activating and fractions near 0 suggest repressive preferences toward the target gene expression. Motif directionality fraction near 0.5 suggests no activity preference or context dependence.

We compared our results to a previously published study that quantified transcription activating or repressive activities based on massively parallel reported assays in HepG2 and K562 cells[34] (Supplementary Fig. 8). We found that the motif directionality measures metric were largely concordant (Spearman's $r = 0.69$, $p = 7.7 \times 10^{-17}$) with orthogonal motif activity measures derived from MPRAs performed in HepG2 and K562 cell line[34] (Supplementary Fig. 8). We then considered 109 motifs from our analyses that were reported to have significant ($p < 0.01$) activating or repressive scores from MPRAs in both HepG2 and K562. With the null expectation of the motif directionality fraction being equal to 0.5, i.e., TF binding equally likely to increase or decrease target gene expression, we used a binomial test to identify TFs that show significant deviation from the null ($N = 18$ at FDR < 10%, Supplementary Data 12).

**Cell culture and transcriptional reporter assays.** MIN6 mouse insulinoma beta-cells[52] were grown in Dulbecco's modified Eagle's Medium (DMEM; Sigma-Aldrich, St. Louis, Missouri/USA) with 10% fetal bovine serum, 1 mM sodium pyruvate, and 0.1 mM beta-mercaptoethanol. INS-1-derived 832/13 rat insulinoma beta-cells (a gift from C. Newgard, Duke University, Durham, North Carolina/USA) were grown in RPMI-1640 medium (Corning, New York/USA) supplemented with 10% fetal bovine serum, 10 mM HEPES, 2 mM L-glutamine, 1 mM sodium pyruvate, and 0.05 mM beta-mercaptoethanol. EndoC-βH1 cells (Endocell) were grown according to Ravassard et al.[49] in DMEM (Sigma-Aldrich), 5.6 mmol/L glucose with 2% BSA fraction V fatty acid free (Roche Diagnostics), 50 μmol/L 2-mercaptoethanol, 10 mmol/L nicotinamide (Calbiochem), 5.5 μg/mL transferrin (Sigma-Aldrich), 6.7 ng/mL selenite (Sigma-Aldrich), 100 U/mL penicillin, and 100 μg/mL streptomycin. Cells were grown on coating consisting of 1% matrigel and 2 μg/mL fibronectin (Sigma). We maintained cell lines at 37 °C and 5% $CO_2$.

To test haplotypes for allele-specific effects on transcriptional activity, we PCR amplified a 765-bp genomic region (element 1) containing variants: rs7798124, rs7798360, and rs7781710, and a second 592-bp genomic region (element 2) containing variants: rs10228796, rs10258074, rs2191348, and rs2191349 from DNA of individuals homozygous for each haplotype. The oligonucleotide primer sequences are listed in Supplementary Table 6. We cloned the PCR amplicons into the multiple cloning site of the Firefly luciferase reporter vector pGL4.23 (Promega, Fitchburg, Wisconsin/USA) in both orientations, as described previously[91]. Vectors are designated as "forward" or "reverse" based on the PCR-amplicon orientation with respect to DGKB gene. We isolated and verified the sequence of five independent clones for each haplotype in each orientation. For the 5′ eQTL a 250 bp construct containing the rs17168486 SNP (Origene) was subcloned into the Firefly luciferase reporter vector pGL4.23 (Promega) in both orientations.

We plated the MIN6 (200,000 cells) or 832/13 (300,000 cells) in 24-well plates 24 h before transfections and plated the EndoC-βH1 cells (140,000 cells) 48 h prior to transfection. We co-transfected the pGL4.23 constructs with phRL-TK Renilla luciferase reporter vector (Promega) in duplicate into MIN6 or 832/13 cells, and in triplicate for EndoC-βH1 cells. For the transfections, we used Lipofectamine LTX (ThermoFisher Scientific, Waltham, Massachusetts/USA) with 250 ng of plasmid DNA and 80 ng Renilla for MIN6 cells, Fugene6 (Promega) with 720 ng of plasmid, and 80 ng Renilla for 832/13 cells per each welll and Fugene6 with 700 ng plasmid and 10 ng renilla for EndoC-βH1 cells. We incubated the transfected cells at 37 °C with 5% $CO_2$ for 48 h. We measured the luciferase activity with cell lysates using the Dual-Luciferase® Reporter Assay System (Promega). We normalized Firefly luciferase activity to the Renilla luciferase activity. We compared differences between the haplotypes using unpaired two-sided $t$-tests. All experiments were independently repeated on a second day and yielded comparable results.

**Electrophoretic mobility shift assays.** EMSAs were performed as previously described. We annealed 17-nucleotide biotinylated complementary oligonucleotides (Integrated DNA Technologies) centered on variants: rs10228796, rs10258074, rs2191348, and rs2191349 (Supplementary Table 7). MIN6 nuclear protein extract was prepared using the NE-PER kit (Thermo Scientific). To conduct the EMSA binding reactions, we used the LightShift Chemiluminescent EMSA kit (Thermo Scientific) following the manufacturer's protocol. Each reaction consisted of 1 μg poly (dI-dC), 1× binding buffer, 10 μg MIN6 nuclear extract, and 400 fmol biotinylated oligonucleotide. We resolved DNA–protein complexes on nondenaturing DNA retardation gels (Invitrogen) in 0.5× TBE. We transferred the complexes to Biodyne B Nylon membranes (Pall Corporation), and UV cross-linked (Stratagene) to the membrane. We used chemiluminescence to detect the DNA–protein complexes. EMSAs were repeated on a second day with comparable results.

**Reporting summary.** Further information on research design is available in the Nature Research Reporting Summary linked to this article.

## Data availability

All relevant data supporting the key findings of this study are available within the article and its Supplementary Information files or from the corresponding author upon

reasonable request. Genotype, technical and biological covariates, and sequence data have been deposited at the European Genome-phenome Archive (EGA; https://www.ebi.ac.uk/ega/) under the following accession numbers: EGAD00001006149; EGAS00001004042; EGAS00001004056. Complete summary statistics for eQTL associations are accessible in the following link: https://zenodo.org/record/3408356. In addition, Source data are provided with this paper.

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

## Acknowledgements

A.Vi. and E.T.D. were supported by EU IMI program (UE7-DIRECT-115317-1), NIH (NIH-R01-MH101814), and FNS funded project RNA1 (31003A_149984). A.Va. was supported by the American Association for University Women International Doctoral Fellowship, Barbour Doctoral Scholarship, and the University of Michigan Rackham Predoctoral Fellowship. M.v.d.B. was supported by a Novo Nordisk postdoctoral fellowship run in partnership with the University of Oxford. R.B.P. was supported by the EFSD/Novo Nordisk Programme for Diabetes Research in Europe, Diabetes Wellness (720-858-16 JDWG), Åke Wiberg Foundation (M18-0216). L.G. was supported by the Swedish Research Council project grant (2015-2558) Swedish Research Council, Astra Zeneca (10033731), Strategic Research Area Exodiab, Dnr 2009-1039, Swedish Foundation for Strategic Research Dnr IRC15-0067, and the Swedish Research Council, Linnaeus grant, Dnr 349-2006-237. F.S.C., M.R.E., and N.N. were supported by NHGRI-ZIA HG000024. A.K.I., S.V., and K.L.M. were supported by NIH R01 DK072193 and NIH U01 DK105561. S.C.J.P., L.J.S., and M.B. were supported by U01DK062370. M.L.S. was supported by K99/R00DK092251. P.O. was supported by grant T32 HG00040 from the National Human Genome Research Institute of the NIH. P.E.M. was supported by a Foundation grant from the Canadian Institutes of Health Research (CIHR: 148451). S.C.J.P. was supported by National Institute of Diabetes and Digestive and Kidney Diseases grants R00 DK-099240 and R01 DK-117960, American Diabetes Association Pathway to Stop Diabetes grant 1–14-INI-7. The Alberta Diabetes Institute IsletCore was supported by the Alberta Diabetes Foundation. We thank the Human Organ Procurement and Exchange (HOPE) program and the Trillium Gift of Life Network (TGLN) for their efforts in obtaining human organs for research. A.L.G. is a Wellcome Senior Fellow in Basic Biomedical Science. This work was funded in Oxford by the Wellcome Trust (095101, 200837, 106130, 203141, Medical Research Council (MR/L020149/1), European Union Horizon 2020 Programme (T2D Systems), NIH (U01-DK105535; U01-DK085545) and by the EU IMI program (UE7-DIRECT-115317-1). M.I.Mc.C was a Wellcome Senior Investigator and an NIHR Senior Investigator. He was supported by the Wellcome Trust (grants nos. 090532, 106130, 098381, 203141, and 212259); Medical Research Council grant no. MR/L020149/1; NIDDK (U01-DK105535, R01-MH101814, and R01-MH090941); NIHR (NF-SI-0617-10090). This work was also supported by the Oxford NIHR Biomedical Research Centre. The views expressed in this article are those of the author(s) and not necessarily those of the NHS, the NIHR, or the Department of Health.

## Author contributions

A.Vi. and C.H. performed the re-mapping, quantification, and quality checks for the join RNA-Seq dataset with assistance from M.v.d.B., J.F., and N.O. M.v.d.B. performed genotype quality evaluation and imputation of the join genotypes data with the assistance of A.M. Geneva samples (GEN) were collected, processed, and quality checked by A.Vi., C.H., N.I.P., A.A.B., and E.T.D. Lund samples (LUN) were collected processed and quality check by R.B.P., O.A., J.F., O.H., G.H., U.K., N.O., and L.G. Oxford and Edmonton samples (OXF) were collected processed and quality checked by M.v.d.B., A.B., P.J., P.E.M., A.M., J.E.M.F., V.N., A.P., A.L.G., and M.I.M. USA samples (USA) as well as ATAC-seq were collected processed and quality checked by A.Va., M.B., M.R.E., N.N., P.O., M.L.S., R.W., F.S.C., L.J.S., and S.C.J.P. Data analyses were performed by A.Vi., A.Va., M.v.d.B., R.B.P., A.A.B., J.F., N.O., A.P., and L.J. Experiments and analyses associated to the validation of *DGKB* eQTLs were performed by V.N., S.V., A.K.L., K.L.M., A.L.G., A.Va., and S.C.J.P. The manuscript was drafted by A.Vi., A.Va., M.v.d.B., L.J.S., S.C.JP., and M.I.M., then revised and approved by all authors.

## Competing interests

M.I.M. has served on advisory panels for Pfizer, Novo Nordisk, and Zoe Global; received honoraria from Merck, Pfizer, Novo Nordisk, and Eli Lilly; and received research funding from Abbvie, Astra Zeneca, Boehringer Ingelheim, Eli Lilly, Janssen, Merck, Novo

Nordisk, Pfizer, Roche, Sanofi Aventis, Servier, and Takeda. As of June 2019, he is an employee of Genentech, and a holder of Roche stock. M.v.d.B. is an employee of Novo Nordisk A/S, although all experimental work was carried out under employment at the University of Oxford.

## Additional information

Ana Viñuela [1,2,3,4,22✉], Arushi Varshney [5,22], Martijn van de Bunt[6,7,8,22], Rashmi B. Prasad [9,22], Olof Asplund[9], Amanda Bennett [6], Michael Boehnke [10], Andrew A. Brown[1,2,3,11], Michael R. Erdos [12], João Fadista[9,13,14], Ola Hansson [9,14], Gad Hatem[9], Cédric Howald[1,2,3], Apoorva K. Iyengar[15], Paul Johnson[6], Ulrika Krus[9], Patrick E. MacDonald[16], Anubha Mahajan [6,21], Jocelyn E. Manning Fox[16], Narisu Narisu [12], Vibe Nylander[7], Peter Orchard[17], Nikolay Oskolkov [9], Nikolaos I. Panousis[1,2,3], Anthony Payne[6], Michael L. Stitzel [18,19], Swarooparani Vadlamudi[15], Ryan Welch [10], Francis S. Collins [12], Karen L. Mohlke [15], Anna L. Gloyn [6,7,8,20], Laura J. Scott [10], Emmanouil T. Dermitzakis [1,2,3], Leif Groop [9,14], Stephen C. J. Parker[5,17] & Mark I. McCarthy [6,7,8,21✉]

[1]Department of Genetic Medicine and Development, University of Geneva Medical School, 1211 Geneva, Switzerland. [2]Institute for Genetics and Genomics in Geneva (iGE3), University of Geneva, 1211 Geneva, Switzerland. [3]Swiss Institute of Bioinformatics, 1211 Geneva, Switzerland. [4]Biosciences Institute, Faculty of Medical Sciences, Newcastle University, NE1 4EP Newcastle, UK. [5]Department of Human Genetics, University of Michigan, Ann Arbor, MI 48109, USA. [6]Wellcome Centre for Human Genetics, Nuffield Department of Medicine, University of Oxford, Oxford OX3 7BN, UK. [7]Oxford Centre for Diabetes, Endocrinology and Metabolism, Radcliffe Department of Medicine, University of Oxford, Oxford OX3 7LE, UK. [8]Oxford NIHR Biomedical Research Centre, Oxford University Hospitals Trust, Oxford OX3 7LE, UK. [9]Lund University Diabetes Centre, Department of Clinical Sciences, Lund University, Skåne University Hospital, Malmö, Sweden. [10]Department of Biostatistics and Center for Statistical Genetics, University of Michigan, Ann Arbor, MI 48109, USA. [11]Population Health and Genomics, University of Dundee, Dundee, Scotland DD1 9SY, UK. [12]Medical Genomics and Metabolic Genetics Branch, National Human Genome Research Institute, National Institutes of Health, Bethesda, MD 20892, USA. [13]Department of Epidemiology Research, Statens Serum Institut, Copenhagen, DK 2300, Denmark. [14]Finnish Institute for Molecular Medicine (FIMM), University of Helsinki, Helsinki, Finland. [15]Department of Genetics, University of North Carolina, Chapel Hill, NC 27599, USA. [16]Department of Pharmacology and Alberta Diabetes Institute, University of Alberta, Edmonton, Alberta, Canada. [17]Department of Computational Medicine & Bioinformatics, University of Michigan, Ann Arbor, MI 48109, USA. [18]The Jackson Laboratory for Genomic Medicine, Farmington, CT 06032, USA. [19]Department of Genetics and Genome Sciences, Institute for Systems Genomics, University of Connecticut, Farmington, CT 06032, USA. [20]Department of Pediatrics, Division of Endocrinology, Stanford School of Medicine, Stanford University, Stanford, CA, USA. [21]Present address: Human Genetics, Genentech, 1 DNA Way, South San Francisco, CA 94080, USA. [22]These authors contributed equally: Ana Viñuela, Arushi Varshney, Martijn van de Bunt, Rashmi B. Prasad. ✉email: ana.vinuela@newcastle.ac.uk; mccarthy.mark@gene.com

