## [Peer Review File · Nature Communications]

Peer Review Information

Manuscript title: Genetic variant effects on gene expression in human pancreatic islets and their implications for T2D

Corresponding author name(s): Dr Ana Vinuela and Prof. Mark I. McCarthy

Editorial notes:

none

Reviewer comments & decisions:

Reviewer comments, first version:

Reviewers' comments:

Reviewer #1 (Remarks to the Author):

1. For the purposes of enabling replication and to allow other researchers to use these valuable data sets in future, the authors should provide a spreadsheet listing all relevant donor characteristics that are typically reported in studies using human islets (donor ID, age, sex, BMI, HbA1c, diabetes status, diabetes duration, cause of death, cold ischemia time, islet prep purity, % viability).

Note: Donor information, and genotyping and sequencing data files need to be labelled in such a manner as to: (1) allow other researchers to link genotyping and islet RNA-seq data with the respective donor information; (2) enable traceability = link each donor ID in this study with the donor ID originally assigned by the islet isolation facility/islet distribution programme where the islets were isolated.

Point (2) is important as centres such as IsletCore and Nordic Islets perform quality control and functional assays on each islet prep and make these data available to reserachers. It is essential to be able to link such data with the genotyping and RNA-seq data provided in this study.

In the same spreadsheet, the authors could also add the results of their "cell deconvolution analysis" shown in Figure SF3 (i.e. proportion of endocrine islet tissue, and estimated % of beta cells and non-beta cells within the islet endocrine fraction).

2. Can the authors comment on the power of this study, given the sample size, to identify a variant as an eQTL, depending on MAF? Also, Huang et al. (2018, Nucleic Acids Research) describe an adapted bootstrap method to correct for Winner's Curse in eQTL studies.

3. Overlap between gene-eQTL and exon-eQTL? The authors mention that “exon-level analysis [...] can capture the impact of variants influencing splicing”, but do not then comment on the findings or report on any specific variants that influence splicing. It would be useful if the authors could provide a table of exon-eQTLs that are not detected in the gene-eQTL analysis, as these are expected to indicate SNPs that affect expression of splice isoforms. One would expect that this list should only include alternatively spliced genes?

4. ATAC-seq data:

a. The authors refer to “ATAC-seq peaks previously identified from two human islet samples” (line 674) but do not include a reference for this data.

b. ATAC-seq data used in the manuscript came from the islets of only 2 donors. This is not due to lack of ATAC-seq data availability. Khetan et al. (Diabetes, 2016) generated ATAC-seq profiles from islets of 19 donors (14 non-diabetic). The preps used by Khetan et al. were of high purity (>80%) and high % viability. The current study’s last author (MIM) and other authors who contributed to this manuscript have themselves published ATAC-seq data from 5 non-diabetic human donor islet preps (Turner et al., Stem Cell Reports, 2017) and then another 17 non-diabetic human donor islet preps (Turner et al., Elife, 2018).

c. Published ATAC-seq data are also available for FAC-sorted beta-cell populations (e.g. Ackermann et al., Mol Metab, 2016).

5. Inevitably, islet samples from the different sources would have been treated differently: they were isolated, handled, cultured, extracted and sequenced in different centres each using a different protocol. The authors mention that they controlled for differences in islet handling across the different study sites, but no further details are provided. It would be useful if the authors could elaborate on how these complex variables were controlled for.

6. Islet isolation and ex vivo culture subjects the tissues to many stresses. Have the authors checked and/or controlled for markers of stress? For example, a simple test would be to check for activation of well-known inflammatory and stress response pathways.

7. Did the authors adjust for the following: cause of death, cold ischemia time, age, gender, BMI, race? Of these, only gender is mentioned in the manuscript. For low MAF SNPs, it is easy to see how a skew in the donor population can lead to an apparent correlation between SNP and gene expression. Race is particularly important here since MAFs will vary between ethnic groups.

8. Most samples have high levels of contamination with exocrine. One possible concern is that many of the identified correlations between variant and gene expression are driven by the impure preps with high contamination of exocrine? The authors did look at the “impact of variation in purity between samples” and report that “more than 100 PCs were required to remove at least 50% of the variance. This indicates that some of the eQTLs here attributed to pancreatic islets may, in fact, reflect exocrine pancreatic contamination.”

The authors use the argument that the overlap between GTEX whole pancreas eQTLs and the islet eQTLs identified in this study is not great, suggesting that this study has identified genuine islet eQTLs

that would be missed in whole pancreas. The authors also suggest that this relatively “poor” overlap is not due to low pancreas numbers in the GTEX data set, as other (unrelated) tissues with larger numbers can also show “poor” overlap. However, data presented in Figure 1A of the manuscript do show that there is a clear correlation between GTEX sample size and overlap with GTEX data, albeit not a linear one. One wonders how much higher the overlap between islet eQTLs identified here and whole pancreas GTEX eQTLs would be if GTEX had data from the pancreata of 400-500 donors instead of <150?

Furthermore, islet purity has been reported to affect in vitro islet function, suggesting that the % of exocrine contamination will change beta-cell gene expression (perhaps due to digestive enzymes or other factors secreted by the exocrine pancreas).

9. Are the eQTLs identified here more truly likely to lie within islet ATAC-seq peaks, TF motifs, and certain chromatin states compared to non-eQTL variants? To suggest this, the authors should present what % of the SNPs analysed, or proxy SNPs, that lie within 1 MB of genes expressed in pancreatic islets would overlap islet ATAC-seq peaks or TF motifs? This analysis would need to use variants within 1 MB of islet-expressed genes to be meaningful, rather than the entire data set, as islet eQTLs can only be identified for islet-expressed genes.

10. For eSNPs that colocalise with glycaemic traits, the authors use this as an argument in support of a role for pancreatic islet dysfunction in leading to dysglycaemia. An alternative explanation could be that a SNP of interest causes dysglycaemia through an effect on another tissue (e.g.: liver) and the altered blood glucose concentration in turn affects islet gene expression.

a. Can the authors distinguish between these two scenarios? One option would be to calculate the overlap between eGenes whose eSNPs overlap with glycaemic trait GWAS signals, and genes whose expression is regulated by glucose in vitro (ref 47 in the manuscript).

b. It would be ideal if the authors could adjust for HbA1c levels in their eQTL analyses.

11. Functional validation, Figure 3F: an effect was only observed in mouse and rat beta-cell lines but not in the EndoC-betaH1 human beta-cell line.

Minor comments:

1. Given the complex design of the study and the large number of different bioinformatic pipelines used, it would be helpful if the authors included panels with diagrams of the experimental design/data processing steps employed to generate the different sets of data. Otherwise, it is rather hard to follow.

2. Line 1143: os \diamond of

3. No journal info is listed for reference 74. It appears that this tool has only been published on BioRxiv in 2015 but has not been published in a peer-reviewed journal in the meantime. Did the authors mean to cite Brown et al. (2015, eLife), instead?

4. Supplementary figure SF7: legend reads “A: Number of eQTL Islet eQTL overlapping with Islet chromatin states and stretch/typical enhancers.” \diamond A: Number of eQTL Islet eQTL overlapping with Islet chromatin states and stretch/typical enhancers.

5. "803 Data access

Genotype and sequence data have been deposited at the European Genome-phenome Archive (EGA; <https://www.ebi.ac.uk/ega/>) under the accession number EGAXXX (submission in process)."

Have the data now been submitted?

Reviewer #2 (Remarks to the Author):

Vinuella et al present the largest eQTL study performed in pancreatic islets, comprising data from 420 donors from several studies.

The main findings are: (a), they demonstrate that some eQTLs are not identifiable in other tissues such as GTEx, indicating the importance of performing eQTLs in the disease target tissue (i. e. pancreatic islets for type 2 diabetes); (b) they show (expected) overlap between epigenome marks in pancreatic islets; (c) enrichment of eQTLs in variants implicated in islet dysfunction; (d) colocalization of islet eQTLs influencing T2D or related trait; (e) effector transcripts for 23 loci.

This is an exceptional resource for those working on genetics and genomics of type 2 diabetes. However, in my opinion, the paper fails to reflect the huge relevance that this study has.

Some major points include the following:

- It is not clear to me, how this manuscript improves the discovery compared to other smaller eQTL studies that have been published? For example, does larger sample size improve the discovery of low-frequency variants associated with expression in eQTLs?
- The authors provide effector transcripts for 23 loci. Given that there exist around ~200 loci in Europeans, and the general believe that most of GWAS loci are driven by islet dysfunction, 23 seems a small percentage. How many of these 23 effector transcripts would not be found in GTEx or have not been found in previous eQTL studies?
- Data sharing. The authors share the significant eQTL results, but I didn't see in the manuscript any sign of how are the authors going to share the eQTL full summary statistics. Since this is such a valuable dataset, I think the authors should at least provide the full summary statistics of all the eQTL gene-pairs, such as GTEx does in its website. dbGaP or EGA access to raw genotype data RNA-seq data is also encouraged, but summary statistics of the association should be provided even if dbGaP access is provided so that investigators interested in a lookup do not have to re-invent the wheel and perform the whole eQTL meta-analysis again.
- Tissue specific regulatory variation in islets. The authors use p-value enrichment between the eQTLs identified in pancreatic islets and other tissues. The authors show that there is a positive linear relationship between the sample size of each tissue, and the p-value enrichment. They then claim that pancreas does not seem a good surrogate. However, when seeing figure 1 it seems to me that pancreas is one of the tissues that shows higher p value enrichment compared to others of similar sample size. The authors claim: "This does not reflect low sample size: the number of whole pancreas

samples is on a par with other tissues such as skin and spleen with comparable eQTL-sharing (r^2 0.67, 0.61 respectively).” I disagree with this statement as skin and spleen have way different sample sizes and they both have lower r^2 . Actually, the skin that shows has actually close to 300 samples, so almost double than Pancreas. Since I believe that this is a very relevant question, could the authors develop a new method for tissue sharing that does not depend us much on sample sizes, i. e. based on effect sizes, rather than sample sizes? Would, for example cross-phenotype LD-score regression type of analysis be suitable to assess that?

- The study of cellular heterogeneity is quite interesting. They provide 18 islet cis-eQTL that are dependent on genotype-by-beta-cell proportion. Did the authors use this cellular proportion as a covariate for the eQTL analysis?

- Overlap with T2D and glycemic GWAS variants. I was a bit surprised that only 78 lead GWAS SNPs from Fuchsberger et al (2016). There have been a number of larger GWAS studies, based on individuals of European ancestry, many of them lead by the same senior author of this study. Why did the authors not use any of the more updated version of GWAS hits? Despite that, the authors show that Glycemic T2D variants and T2D (beta-cell cluster) variants, only show enrichment in islet, but not in Pancreas or any other tissue. This is, in my opinion, the most relevant result and what convinced me that eQTLs in pancreatic islets are important. Perhaps mentioning that in the abstract is important.

- Functional validation of DGKB locus. I do not have a lot of experience in EMSA, but I believe that this experimental validation and its interpretation the should be explained more in detail as it is hard to follow.

- Overall, I feel that the writing could be less verbose and more specific, as some of the paragraphs are difficult to follow. I would recommend to thoroughly review the text to improve clarity and consistency.

Methods: “Quality of genotyping from the shared SNPs in both arrays was assessed before imputation separately by removing SNPs as follows” Does that mean that only the overlapping SNPs were used before imputation? What was the number of overlapping SNPs between these two panels?

- Where all the samples imputed together? Or by cohort, or by panel?

Minor comments:

- The naming of the suppl are difficult to follow, as the name does not correspond to the suppl table number. Also, the tables do not have caption, which would be appreciated. So I still do not know which table is which.

- I miss in the discussion a section that tells out how much improvement was gained by augmenting the sample size. How much larger was the number of eQTLs identified? Was there an improvement in identifying eQTLs driven by low-frequency variants? How much did the number of candidate effector genes improve?

Reviewer #3 (Remarks to the Author):

This manuscript describes a large eQTL dataset generated in pancreatic islets, the first such dataset I think that exists. It will be a useful resource for many researchers and the description of it in the manuscript is clear. The demonstration that cell type specificity of effects is important, and fits with other data in this area, and the match of cell type eQTL to previously assigned GWAS categories is a nice result.

I have only minor comments:

I was not familiar with the notion of p value enrichment analysis, nor the technique to perform it (p4). I had to go read the referenced paper, and I think it would be helpful if the authors were to include a brief outline of what the method intends to achieve, and how, before presenting their results.

On page 5/6 eSNPs within stretch enhancers are described as having "smaller effects" (last line on p5). Then (top of p6) such eSNPs are described as requiring larger sample sizes for "equivalent effect size". This is confusing and needs clarifying. Presumably the two "effect size" used mean different things here (eg fold change vs variance explained?).

When this effect is discussed later (p12) it is suggested that this means that a GWAS causal variant sitting in an islet stretch enhancer could be misassigned to a non-islet expressed gene because its effect on islets differs from its effect on bystander genes in other tissues. I don't think this is shown - the effects of stretch enhancer SNPs to non stretch enhancer SNPs is compared within islets, but the effects of islet stretch enhancer SNPs on islet genes vs non-islet genes (where the stretch enhancer is not operating as a stretch enhancer??) are not compared. Either I have misunderstood this argument and it needs making clearer, or additional data are needed to support the claim.

p9, discussing colocalisation of PDE8B, the authors describe the existence of two signals, which violates the assumption of the coloc approach. They have the data, and can condition on one of the two signals for each trait to test for colocalisation of the "other" signal (so 4 tests in total). If only summary data are available, COJO enables this.

I would encourage the authors to complete the deposition of the data in EGA, and editors to confirm deposition is complete, because the sharing of this dataset will enable the widest utility for the work. (These authors have a previously very good record of sharing data - I'm saying this only because previous papers I have reviewed by other authors with "deposition in progress" have not always resulted in an actually deposited dataset).

Author rebuttal, first version:**Reviewer #1 (Remarks to the Author):**

1. For the purposes of enabling replication and to allow other researchers to use these valuable data sets in future, the authors should provide a spreadsheet listing all relevant donor characteristics that are typically reported in studies using human islets (donor ID, age, sex, BMI, HbA1c, diabetes status, diabetes duration, cause of death, cold ischemia time, islet prep purity, % viability).

Note: Donor information, and genotyping and sequencing data files need to be labelled in such a manner as to: (1) allow other researchers to link genotyping and islet RNA-seq data with the respective donor information; (2) enable traceability = link each donor ID in this study with the donor ID originally assigned by the islet isolation facility/islet distribution programme where the islets were isolated.

Point (2) is important as centres such as IsletCore and Nordic Islets perform quality control and functional assays on each islet prep and make these data available to researchers. It is essential to be able to link such data with the genotyping and RNA-seq data provided in this study.

In the same spreadsheet, the authors could also add the results of their “cell deconvolution analysis” shown in Figure SF3 (i.e. proportion of endocrine islet tissue, and estimated % of beta cells and non-beta cells within the islet endocrine fraction).

We agree with both the reviewer’s points and the objectives of the recent proposal to share relevant biological information from donors. However, many of the samples included in the present analysis predate those recommendations by several years, and the relevant data were not collected, or are inaccessible due to ethical constraints that are designed to preclude donor identification. We are happy to provide the limited data that we have, and this has been included with the EGA submission as covariate information and linked to the sample IDs used for genotype and expression data. These files contain principal components (PCs) from expression, PCs from genotypes and the proportion of cell types used in the analyses in addition to complete information about sex and T2D status. We also included the information we have regarding age (11 missing), BMI (87 missing), % of HbA1c (222 missing), islet viability (394 missing) and islet purity (166 missing). In addition, we have now included the following summary of the available information in the methods section (Page 13):

The samples from 420 donors included 189 males and 231 females, with an age range of 16 to 81 years (median = 54 years, 11 not available (NAs)). Of the 420 individuals, 37 were identified as diabetic. BMI information was available for 334 individuals (median, 26.3kgm⁻²), while HbA1c measurements were available for 198 (median, 5.8%). Due to the historical nature of some of the samples used in this study, QC information about the pancreatic islet isolation was limited: 254 samples listed their purity (median, 75%) and 26 samples listed their viability (median, 93.5%). No other biological information was available in the historical records. This information is included in the covariate files included in the EGA submission.

2. Can the authors comment on the power of this study, given the sample size, to identify a variant as an eQTL, depending on MAF? Also, Huang et al. (2018, Nucleic Acids Research) describe an adapted bootstrap method to correct for Winner's Curse in eQTL studies.

All eQTL studies are limited by sample size: however this is the best powered RNAseq study in pancreatic islets to date. Ongoing efforts to gather more samples will undoubtedly increase the potential for discovery both of weaker effects (for common variants), but also enhance the potential to uncover eQTLs arising from low frequency variants. In the current analysis, we limit our analyses to SNPs with MAF>1%.

In relation to the reviewer's comment about a "winner's curse" issue in eQTL studies, as we understand it, the method proposed by Huang et al. was designed to evaluate studies with much smaller sample sizes, where this may be a significant issue. Given that our sample size appreciably exceeds those for which the approach was designed, there seems to be no basis for reanalysis using the method proposed.

3. Overlap between gene-eQTL and exon-eQTL? The authors mention that "exon-level analysis [...] can capture the impact of variants influencing splicing", but do not then comment on the findings or report on any specific variants that influence splicing. It would be useful if the authors could provide a table of exon-eQTLs that are not detected in the gene-eQTL analysis, as these are expected to indicate SNPs that affect expression of splice isoforms. One would expect that this list should only include alternatively spliced genes?

We have now included two lists with exon-eQTLs and gene-eQTLs that were not significant in the other analysis. These include 57 genes with gene-eQTLs but no significant exon-eQTLs (Supplemental table 3), and 3,863 genes that had no significant gene-eQTLs but had exon-eQTLs (Supplemental table 4). In our view, the gene- and exon-level analyses are complementary and discrepancies between the two are not the most reliable means to capture splicing effects. These related phenotypes have different properties and slightly different power to identify different types of expression effects. Both phenotypes are expected to include eQTLs associated with changes in absolute expression and with splicing. We agree that, in general, gene level quantifications are more likely to miss splice-eQTLs, as the phenotype used (RPKM) averages the expression of all the exons, and can mask changes in expression that affect only one exon (e.g. exon-skipping events). On the other hand, exon-level quantifications come with a substantial increase in multiple testing burden (N exon phenotypes = 168,833, N gene phenotypes = 22,169), and a SNP altering the expression of the whole gene may not be significant after correction if the overall effect is low. However, it is also possible to think of scenarios where splicing events are only seen in the gene-level RPKM values (e.g. alternative promoter, but small overall change in expression).

4. ATAC-seq data:

a. The authors refer to “ATAC-seq peaks previously identified from two human islet samples” (line 674) but do not include a reference for this data.

We thank the reviewer for pointing this out. We have included the reference on page 6 with the updated list of samples now used (see next point).

b. ATAC-seq data used in the manuscript came from the islets of only 2 donors. This is not due to lack of ATAC-seq data availability. Khetan et al. (Diabetes, 2016) generated ATAC-seq profiles from islets of 19 donors (14 non-diabetic). The preps used by Khetan et al. were of high purity (>80%) and high % viability. The current study’s last author (MIM) and other authors who contributed to this manuscript have themselves published ATAC-seq data from 5 non-diabetic human donor islet preps (Thurner et al., Stem Cell Reports, 2017) and then another 17 non-diabetic human donor islet preps (Thurner et al., Elife, 2018).

We thank the reviewer for their constructive feedback. We downloaded the islet ATAC-seq data (raw fastq files) for 14 non-diabetic donors from Khetan et al. (DOI 10.2337/db18-0393) and 17 non-diabetic donors from Thurner et al. (DOI 10.7554/eLife.31977) and processed these uniformly along with the two non-diabetic samples (DOI 10.1073/pnas.1621192114) included in the initial submission of our manuscript. Since these 33 samples were obtained from different studies, we sought to obtain the set of peaks that were reproducible. We subsampled each ATAC-seq sample bam file to the minimum read depth across samples of 27,994,993 reads, merged these subsampled bam files and called peaks using (1) the merged bam file that uniformly represents each sample and (2) each individual sample bam file. We then selected peaks from the merged bam file that were reproducibly called across the majority (at least 17) of the 33 individual samples, resulting in 64,129 peaks.

We re-ran all analyses in our manuscript involving ATAC-seq data using this new set of reproducible peaks and present the updated results in revised figure 2 (2B, 2C and 2D) and Supp Figs. 12 to 16. These new results are consistent with our previously reported results:

- 1. In agreement with our initial submission, we observe that islet eQTLs show enrichment of overlap with ATAC-seq peaks (fold enrichment = 2.17, $P = 3.78 \times 10^{-206}$), and that effect sizes of eQTLs occurring in ATAC-seq peaks in stretch enhancer states are lower than those occurring in ATAC-seq peaks in the active TSS chromatin states ($P = 0.0034$).*

2. *For analyses involving TF footprint motifs, we originally utilized the union of footprint motifs across two islet samples. We have now subsetted the footprint motifs selecting occurrences within the new set of reproducible ATAC-seq peaks. We again observe that overlaying eQTL data with ATAC-seq-informed TF footprint motifs reveals in vivo motif directionalities (Figure 2D).*
3. *Twenty-three motifs pass the 10% FDR threshold after the binomial test for motif directionality for significant deviation from 0.5, which is more than we observed previously (N = 8 significantly deviated at 10% FDR).*
4. *Additionally, the correlation of motif directionality with MPRA data increased (previous Spearman's $R = 0.64$, $p = 8.1 \times 10^{-13}$; updated Spearman's $R = 0.73$, $p = 1.5 \times 10^{-17}$) when using the new subsetted peak results.*

Collectively, these results indicate that the reproducible peaks identified from 33 islet ATAC-seq datasets made our analyses more robust. We thank the reviewer for this recommendation, which has strengthened our manuscript.

c. Published ATAC-seq data are also available for FAC-sorted beta-cell populations (e.g. Ackermann et al., Mol Metab, 2016).

We thank the reviewer for this suggestion. Here, we reasoned that because we utilize ATAC-seq data along with eQTLs which were identified using RNA-seq from bulk islet tissue, ATAC-seq data obtained from bulk islet samples would be more relevant than data from FAC-sorted beta cells. For these reasons, we have followed feedback from comment 4.b above and included bulk islet ATAC-seq data from 33 non-diabetic donors.

5. Inevitably, islet samples from the different sources would have been treated differently: they were isolated, handled, cultured, extracted and sequenced in different centres each using a different protocol. The authors mention that they controlled for differences in islet handling across the different study sites, but no further details are provided. It would be useful if the authors could elaborate on how these complex variables were controlled for.

We apologize for the oversight in this point and we have now expanded the text describing how this is done (METHODS: "RNAseq quality assessments and data normalization", Page 15 main text, and Page 7 Supplemental methods note). In short, all the differences are grouped in a variable identifying the lab of origin of the samples. Since, as the reviewer pointed out, each institution handled samples in a different way with different processing and sequencing protocols, such a variable will capture all those

differences. In addition, and to control for global and unknown variables that may influence a subset of samples not controlled using a "lab-level" variable, we used principal component analysis and selected the first 25 principal components of expression (PCs) as covariates for the eQTL analysis, a standard, and widely-used approach for eQTL analyses. Supplemental figure 1 shows that the comparison of PC1 and PC2 can successfully identify batch effects internal to samples from one specific lab e.g. for OXF samples and LUN samples. Since cis-eQTLs have a local effect in expression, controlling for PCs as covariates in a linear model removes unwanted global variation not relevant for cis-genetic effects in expression. In addition, and to control for potential population stratification effects and sex differences, we included 3 PCs derived from the genotypes summarizing the genetic variability of the samples and sex. Finally, we performed a further permutation test where samples were permuted within each lab to preserve the lab effect: with 25 PCs, eQTL discovery was flat, meaning that the lab differences were not generating false positive eQTLs.

6. Islet isolation and ex vivo culture subjects the tissues to many stresses. Have the authors checked and/or controlled for markers of stress? For example, a simple test would be to check for activation of well-known inflammatory and stress response pathways.

There was no specific control for markers of stress, but our expectation is that differences in stress levels are global effects affecting multiple genes, and as such this is precisely the kind of effect that can be captured and removed by controlling for PCs (see above). To evaluate this specific point, we examined the most informative 500 exons from each of the first 5 PCs of expression, which explain 56.9% of the variance in expression in the whole dataset. We then investigated if there was any functional enrichment of these top genes associated to stress. For most PCs, the top genes are associated with phosphorylation and signal transduction, with PC3 showing enrichment for genes associated to apoptosis (KEGG pathway, p-value enrichment $1.5e-5$), all of which may be considered stress signals (Nadal et al, Nat Gen. Rev., 2011) These results supports our expectations that with PCs as covariates we controlled, at least partially, for stress induced responses that may differ across samples.

7. Did the authors adjust for the following: cause of death, cold ischemia time, age, gender, BMI, race? Of these, only gender is mentioned in the manuscript. For low MAF SNPs, it is easy to see how a skew in the donor population can lead to an apparent correlation between SNP and gene expression. Race is particularly important here since MAFs will vary between ethnic groups.

As the information for most of those potential co-founders was not available due to the historical nature of the samples, direct adjustment for these factors was not possible. In such circumstances, where relevant covariates may not be fully measured (or even measured at all), principal component analysis provides the most appropriate (and widely used) solution. By using PCs as covariates we control for any

factors, measured or not, that might have a global impact on expression, and which might therefore increase the risk of false positives by artificially grouping samples based on specific factor exposures. In addition, we included 3 PCs from the genotypes, controlling therefore for the specific case possibility of population structure effects affecting our analysis.

8. Most samples have high levels of contamination with exocrine. One possible concern is that many of the identified correlations between variant and gene expression are driven by the impure preps with high contamination of exocrine? The authors did look at the “impact of variation in purity between samples” and report that “more than 100 PCs were required to remove at least 50% of the variance. This indicates that some of the eQTLs here attributed to pancreatic islets may, in fact, reflect exocrine pancreatic contamination.”

The authors use the argument that the overlap between GTEx whole pancreas eQTLs and the islet eQTLs identified in this study is not great, suggesting that this study has identified genuine islet eQTLs that would be missed in whole pancreas. The authors also suggest that this relatively “poor” overlap is not due to low pancreas numbers in the GTEx data set, as other (unrelated) tissues with larger numbers can also show “poor” overlap. However, data presented in Figure 1A of the manuscript do show that there is a clear correlation between GTEx sample size and overlap with GTEx data, albeit not a linear one. One wonders how much higher the overlap between islet eQTLs identified here and whole pancreas GTEx eQTLs would be if GTEx had data from the pancreata of 400-500 donors instead of <150?

Furthermore, islet purity has been reported to affect in vitro islet function, suggesting that the % of exocrine contamination will change beta-cell gene expression (perhaps due to digestive enzymes or other factors secreted by the exocrine pancreas).

We thank the reviewer for mentioning this point, giving us the chance to clarify our arguments here. Two issues need clarification: the effect of exocrine contamination on the discovery of islet eQTLs, and the influence of sample size on the ability to identify islet eQTLs.

The influence of exocrine contamination is further described in the Supplemental Methods Note (pages 2 and 3), addressing the specific point made by the reviewer. There, we evaluated the replication rate of eQTLs from 100 randomly selected whole pancreas GTEx samples in two sets of results: i) eQTLs derived from 100 islet samples with the largest proportion of exocrine component and ii) 100 islets samples with the lowest proportion of exocrine component. Our analyses showed that samples with higher exocrine contributions show higher degree of similarity ($\rho = 0.75$) with whole pancreas eQTLs than do samples with a lower proportion of exocrine ($\rho = 0.64$). This analysis supports our point that differences

between whole pancreas and islet eQTLs are partially driven by cell-specific effects, while similarities between both are driven by shared genetic effects and exocrine contamination in islet samples.

In relation to the sample size influence on eQTL discovery, current knowledge indicates that with larger eQTL studies, more tissue or cell specific eQTLs are likely to be identified. This is certainly the case for larger multi-tissue studies such as GTEx which are now tending to find lower percentages of shared signals across tissues than previously (when sample sizes were lower). We would expect, by analogy, that with additional samples, the overall percentage of shared signals would decrease. However, we agree with the reviewer that an evaluation of the influence of sample size differences in the discovery of shared genetic signals was missing in the manuscript.

As we cannot increase our sample size, we repeated the eQTL analysis and tissue comparison of eQTLs with a reduced islet dataset to show a fair comparison of whole pancreas eQTLs with islet eQTLs. The figure below shows the same analysis as that presented in Figure 1A when we randomly downsample the number of islet samples to 149 (to match that of GTEx whole pancreas) and compare against data from all the GTEx samples (up to 382 samples). With islet eQTLs calculated from the downsampled islets, we observe that multiple tissues capture high proportions of shared eQTL signals with islets including adipose (0.89), pancreas (0.84), LCLs (0.84) and esophagus (0.84). The overall proportion of islet eQTL signals identified in other tissues increased to a range of 0.53 to 0.89 (compared to the range of 0.40-0.73 seen in the full islet data set). These findings support the expectation that even larger sample size eQTL studies in islets will increase the capacity to detect a higher proportion of tissue or cell specific signals (decreasing even more the range of shared signal across tissues).

Overall, both our results indicate that some tissues are better proxies for non-accessible tissues than other tissues: whole pancreas and adipose are better proxy tissues for islets than skeletal muscle. But we find no support to the notion that, in the absence of the effect of exocrine contamination in the islets, whole pancreas is a better proxy than other tissues.

9. Are the eQTLs identified here more truly likely to lie within islet ATAC-seq peaks, TF motifs, and certain chromatin states compared to non-eQTL variants? To suggest this, the authors should present what % of the SNPs analysed, or proxy SNPs that lie within 1 MB of genes expressed in pancreatic islets would overlap islet ATAC-seq peaks or TF motifs? This analysis would need to use variants within 1 MB of islet-expressed genes to be meaningful, rather than the entire data set, as islet eQTLs can only be identified for islet-expressed genes.

We thank the reviewer for this comment. We present eQTL enrichment results in ATAC-seq peaks, chromatin states and TF footprint motifs performed using the GREGOR tool (DOI 10.1093/bioinformatics/btv201) in Figure 2C and Supp. figures S5 and S7. GREGOR identifies a set of control SNPs based on MAF, distance to the nearest gene and number of SNPs in LD within the provided threshold matched with the input eQTL SNPs. Here, the reviewer raised a good point that for eQTL enrichments, the control SNPs should also be selected from within 1Mb of islet-expressed genes for which eQTLs were tested. Here, we address this question from two complementary perspectives. First, we checked if the control SNPs from the GREGOR method meet the criterion to be around islet expressed genes. We found that:

- *Since the insPIRE dataset includes RNA-seq data for >400 samples, a large number of genes (exon QTL scan included 21,319 unique genes) could be quantified and tested for eQTL.*

- Looking at the set of control SNPs from the GREGOR analysis, we find that out of the total set of 6,031,279 control SNPs, 98.6% (5,949,654) of SNPs occur within 1Mb of TSS for genes for which exon level eQTLs were tested.
- Based on the above two points, we believe our enrichment testing was reasonable because the vast majority (98.6%) of control SNPs were selected from within 1Mb of islet expressed and eQTL tested genes.

Second, to further independently verify our enrichment results, we used a different tool - *fenrich* (DOI 10.1038/ncomms15452) - to test for eQTL enrichments. Importantly, *fenrich* takes into account the space of expression-quantified (tested) genes when calculating enrichments. Using *fenrich*, we still observed eQTL enrichment in chromatin features such as islet ATAC-seq peaks, active TSS and active enhancer chromatin states, and depletion in repressed polycomb chromatin states (see reviewer figure below, error bars represent 95% confidence intervals).

Collectively, these observations indicate that our original enrichment results are robust and not biased by improper proxy SNP selection. We have elaborated on this point in the methods section “Enrichment of genetic variants in genomic features” (Page 19 main text, Page 11 Supplemental methods note).

10. For eSNPs that colocalise with glycaemic traits, the authors use this as an argument in support of a role for pancreatic islet dysfunction in leading to dysglycaemia. An alternative explanation could be that a SNP of interest causes dysglycaemia through an effect on another tissue (e.g.: liver) and the altered blood glucose concentration in turn affects islet gene expression.

The reviewer proposes an alternative route whereby colocalization between a T2D GWAS signal and an islet cis-eQTL is mediated not through islets alone, but through regulatory impacts on other tissues (such as liver or fat). Whilst we agree that this could be contributing at some loci, evidence presented within the manuscript suggests this is not likely to be the dominant explanation for our findings. The strong enrichment for active islet regulatory regions (Figure 3E), and the overlap of islet cis-eQTLs with T2D GWAS signals (which becomes more pronounced in terms of effect size in the subset of loci acting through insulin secretion) indicates that most of the GWAS/islet cis-eQTL overlap is likely to be mediated through direct islet effects.

a. Can the authors distinguish between these two scenarios? One option would be to calculate the overlap between eGenes whose eSNPs overlap with glycaemic trait GWAS signals, and genes whose expression is regulated by glucose in vitro (ref 47 in the manuscript).

We interpreted this reviewer comment as asking whether eGenes co-localizing with glycaemic traits may be mediating their activity through response to glucose stimulation, and not necessarily through direct effects on islet. We had already performed an analysis of the overlap between eGenes for glycaemic traits and genes whose expression is regulated by glucose in vitro in ref 47, and these are provided in supplementary table 21 and page 9 of the manuscript. For example, the eGenes G6PC2, DGKB and GPSM1 are differentially expressed between islets from T2D donors compared to non-T2D. The expression of multiple genes is altered upon glucose treatment of islets from normoglycaemic donors including STARD10, RDH5, WARS, WDR25, SIX2, SIX3, NKX6-3. Furthermore, the expression of SIX3 was also altered after high glucose treatment of islets from type 2 diabetic donors. We also checked MRbase and can confirm that none of the gene-SNP pairs were reported to have causal effects in the liver.

b. It would be ideal if the authors could adjust for HbA1c levels in their eQTL analyses.

Due to the historical origin of the samples, HbA1c levels were only available for 198 individuals, making it impossible to include these data directly in the model. As discussed above, the appropriate and commonly used remedy in such a situation is to use the PC-based approach: we included 25 principal components of expression as covariates to capture global effects without the need for their specific identification. To confirm that we control for potential differences in HbA1c, we performed two tests. First, we evaluated the association of the available measurements with the principal components for expression included in the eQTL analysis. Of the 25 PCs used in the analysis, PCs 16 and 20 were both associated (linear model association with P value < 0.05) with HbA1c levels, indicating we control for global effects related to differences in HbA1c on expression by including PCs of expression as covariates for the eQTL association analysis. Second, we tested for association of all exons with the available HbA1c values after controlling for 25PC of expression, 3 PCs for genotypes, lab of origin, and sex, as we did for the eQTL analysis. Of all exons tested (169,820), 194 (0.11%) were significantly associated with HbA1c at a 1% FDR. Given this low number, we believe possible differences in HbA1c had a negligible effect on the eQTL discovery given the procedures we used.

11. Functional validation, Figure 3F: an effect was only observed in mouse and rat beta-cell lines but not in the EndoC-betaH1 human beta-cell line.

The experiments shown in Figure 3F in three beta cell lines rigorously evaluate whether the candidate haplotype affects transcriptional activity in a direction consistent with the eQTL. These experiments assume that transcription factor binding site motifs are conserved across species, as is frequently observed (PMID 25779349). Notably, we observed the same direction of differences between haplotypes across all three cell lines. Although we observed that the relative luciferase reporter levels were lowest for EndoC-βH1, these cells are also the hardest to grow and transfect and generated the lowest levels of transcriptional activity. We believe the less marked difference observed in EndoC-βH1 cells may be related to the lower level of transcriptional activity observed in these cells, but other explanations are possible.

We revised this text to better describe the results (page 10).

The T2D-risk haplotype showed higher expression than the non-risk haplotype in 832/13 ($P=1.9\times 10^{-4}$) and MIN6 ($P=1.1\times 10^{-6}$) (Figure 3F), which is consistent with the eQTL direction (Figure 3D). Equivalent data for the human EndoC-βH1 cell line was directionally-consistent but not significant (Figure 3F).

Minor comments:

1. Given the complex design of the study and the large number of different bioinformatic pipelines used, it would be helpful if the authors included panels with diagrams of the experimental design/data processing steps employed to generate the different sets of data. Otherwise, it is rather hard to follow.

We have now added diagrams along the supplemental methods notes to the improved descriptions (Supplemental methods note).

2. Line 1143: os of

This has now been corrected in the text.

3. No journal info is listed for reference 74. It appears that this tool has only been published on BioRxiv in 2015 but has not been published in a peer-reviewed journal in the meantime. Did the authors mean to cite Brown et al. (2015, eLife), instead?

The correct citation is used here. The tool cited is a fast implementation of the already existing R package qvalue and while the author felt it had sufficient value to be used by others, he did not consider it warranted a peer-review publication. It is reported here for reproducibility reasons as a full description of the tool is provided in the preprint as well as the link to the source code. The software provides the same results as the qvalue R implementation, which is also cited.

4. Supplementary figure SF7: legend reads “A: Number of eQTL Islet eQTL overlapping with Islet chromatin states and stretch/typical enhancers.” A: Number of eQTL Islet eQTL overlapping with Islet chromatin states and stretch/typical enhancers.

We thank the reviewer for pointing this out. We corrected this in Page 7 of the supplemental figures.

5. “803 Data access

Genotype and sequence data have been deposited at the European Genome-phenome Archive (EGA; <https://www.ebi.ac.uk/ega/>) under the accession number EGAXXX (submission in process).”

Have the data now been submitted?

Yes, all data have now been submitted with the following accession numbers for the different datasets: Genotype, technical and biological covariates, and sequence data have been deposited at the European Genome-phenome Archive (EGA; <https://www.ebi.ac.uk/ega/>) under the following accession numbers: EGAS00001003997; EGAS00001004042; EGAS00001004044; EGAS00001004056. Complete summary statistics for eQTL associations have been deposited and are accessible in the following link: <https://zenodo.org/record/3408356>

Reviewer #2 (Remarks to the Author):

Vinuela et al present the largest eQTL study performed in pancreatic islets, comprising data from 420 donors from several studies.

The main findings are: (a), they demonstrate that some eQTLs are not identifiable in other tissues such as GTEx, indicating the importance of performing eQTLs in the disease target tissue (i. e. pancreatic islets for type 2 diabetes); (b) they show (expected) overlap between epigenome marks in pancreatic islets; (c) enrichment of eQTLs in variants implicated in islet dysfunction; (d) colocalization of islet eQTLs influencing T2D or related trait; (e) effector transcripts for 23 loci.

This is an exceptional resource for those working on genetics and genomics of type 2 diabetes. However, in my opinion, the paper fails to reflect the huge relevance that this study has.

Some major points include the following:

1) It is not clear to me, how this manuscript improves the discovery compared to other smaller eQTL studies that have been published? For example, does larger sample size improve the discovery of low-frequency variants associated with expression in eQTLs?

A larger sample size allows for the discovery of more eQTLs, but more importantly it allowed us to explore independent secondary eQTL signals per gene. Sample size and the analysis of secondary independent signals allowed us to identify 7,741 eQTLs, which is a substantial improvement from the ~4,000 eQTLs previously discovered in Varshney et al, 2017. That study reported eQTLs only from SNPs with $MAF > 5\%$ after performing a meta-analysis, while van de Bunt et al (2015) reported eQTLs results using $MAF > 1\%$, but discovered only 2,341 eQTLs in total. We report here 1,045 eQTLs with lead variants with $1\% > MAF < 5\%$. We chose not to explore variants with MAF below 1% as the sample size precludes robust eQTL identification from such rare SNPs. In addition, in the present study, we also explore issues of tissue-specificity and performed an extensive analysis of functionally relevant annotation for islet eSNPs using the more detailed regulatory annotations now available. All this, together with the updated GWAS results for T2D now presented in the manuscript provides, we believe, a substantial advance in our knowledge the genetic regulation of gene expression in pancreatic islets with implications for our understanding of the genetics of T2D and other glycemetic traits.

2) The authors provide effector transcripts for 23 loci. Given that there exist around ~200 loci in Europeans, and the general believe that most of GWAS loci are driven by islet dysfunction, 23 seems a small percentage. How many of these 23 effector transcripts would not be found in GTEx or have not been found in previous eQTL studies?

We are now reporting in Supplemental table 20 a summary of the previous findings from other eQTL studies regarding co-localization of GWAS variants associated to T2D. In addition and following the recommendations in point 6, we have updated the list of GWAS variants included in the colocalization analysis. This has increased the number of identified effector transcripts to 23 loci (24 signals) with strong evidence and an additional 24 signals supported only by one colocalization method (57 in total). This still represents only around 10% of the total number of T2D and glycemetic trait GWAS signals.

Several factors are likely to be contributing to this. First, whilst most GWAS signals for these traits do appear to operate through changes in insulin secretion, there are many that do not. Second, not all islet-mediated effects will be visible in eQTL studies of basal expression in adult islets (such as those acting during development, or through transcriptional effects only revealed by stimulation). Third, there are issues of power to be considered, especially in the case of transcriptional effects that are mediated through minority cell populations within the islet. Fourth, there are issues related to data processing that

modify the information available for the colocalization analysis (the MTNR1B cis-eQTL is a case in point – see page 8, 4th paragraph). Fifth, we had strict requirements for colocalization and used statistical methods that were not used in previous studies, allowing us to be more confident about our findings. Above and beyond these, it is also possible that cis-effects on expression may not be the primary mechanism of action of some GWAS variants, as highly regulated genes are less likely to show the large transcriptional variations observed in cis effects and may be acting as trans regulators.

3) Data sharing. The authors share the significant eQTL results, but I didn't see in the manuscript any sign of how are the authors going to share the eQTL full summary statistics. Since this is such a valuable dataset, I think the authors should at least provide the full summary statistics of all the eQTL gene-pairs, such as GTEx does in its website. dbGaP or EGA access to raw genotype data RNA-seq data is also encouraged, but summary statistics of the association should be provided even if dbGaP access is provided so that investigators interested in a lookup do not have to re-invent the wheel and perform the whole eQTL meta-analysis again.

All data have now been submitted to EGA with the following accession numbers for the different datasets: Genotype, technical and biological covariates, and sequence data have been deposited at the European Genome-phenome Archive (EGA; <https://www.ebi.ac.uk/ega/>) under the following accession numbers: EGAS00001003997; EGAS00001004042; EGAS00001004044; EGAS00001004056. Complete summary statistics for eQTL associations have been deposited and are accessible in the following link: <https://zenodo.org/record/3408356>

4) Tissue specific regulatory variation in islets. The authors use p-value enrichment between the eQTLs identified in pancreatic islets and other tissues. The authors show that there is a positive linear relationship between the sample size of each tissue, and the p-value enrichment. They then claim that pancreas does not seem a good surrogate. However, when seeing figure 1 it seems to me that pancreas is one of the tissues that shows higher p value enrichment compared to others of similar sample size. The authors claim: "This does not reflect low sample size: the number of whole pancreas samples is on a par with other tissues such as skin and spleen with comparable eQTL-sharing (π_1 0.67, 0.61 respectively)." I disagree with this statement as skin and spleen have way different sample sizes and they both have lower π_1 . Actually, the skin that shows has actually close to 300 samples, so almost double than Pancreas. Since I believe that this is a very relevant question, could the authors develop a new method for tissue sharing that does not depend us much on sample sizes, i. e. based on effect sizes, rather than sample sizes? Would, for example cross-phenotype LD-score regression type of analysis be suitable to assess that?

To create an estimate of enrichment that is not biased by sample size, we limited ourselves to those GTEx tissues with more than 149 samples, randomly chose 149 samples for each of those, and repeated the eQTL mapping. To calculate confidence intervals for our π_1 estimates, we then bootstrapped the eQTL p values 1000 times, shown in the figure below. This analysis found pancreas to be the third most similar tissue (right of the plot), after adipose visceral and aorta, but the wide overlapping confidence intervals mean that these tissue differences were not significant. In addition, it is possible that exocrine contamination of islet preparations biases our similarity estimates towards pancreas and islets being more similar than they would be had we pure samples of each (see supplementary methods note, page 2 and 3).

Therefore, we stand by our conclusions that there is no evidence to suggest that whole pancreas is a better proxy tissue for islets than other tissues that show similar degree of sharing.

Figure showing π_1 enrichment analysis with confidence intervals with eQTL comparisons from downsized samples ($n < 149$).

5) The study of cellular heterogeneity is quite interesting. They provide 18 islet cis-eQTL that are dependent on genotype-by-beta-cell proportion. Did the authors use this cellular proportion as a covariate for the eQTL analysis?

No, we control using PCs for expression and genotype, which is currently the standard approach used by the field for eQTL mapping.

6) Overlap with T2D and glycemic GWAS variants. I was a bit surprised that only 78 lead GWAS SNPs from Fuchsberger et al (2016). There have been a number of larger GWAS studies, based on individuals of European ancestry, many of them lead by the same senior author of this study. Why did the authors not use any of the more updated version of GWAS hits? Despite that, the authors show that Glycemic T2D variants and T2D (beta-cell cluster) variants, only show enrichment in islet, but not in Pancreas or any other tissue. This is, in my opinion, the most relevant result and what convinced me that eQTLs in pancreatic islets are important. Perhaps mentioning that in the abstract is important.

We used the opportunity, whilst preparing this revision, to update our analyses to accommodate the larger number of GWAS signals that have become available since our original analysis. These results are now described on pages 8 and 9.

7) Functional validation of DGKB locus. I do not have a lot of experience in EMSA, but I believe that this experimental validation and its interpretation should be explained more in detail as it is hard to follow.

We have revised the text to clarify the different experiments performed to assess the functional role of eSNPs associated to DGKB (Page 9 to 11, and Supplemental methods page 13 and 14).

8) Overall, I feel that the writing could be less verbose and more specific, as some of the paragraphs are difficult to follow. I would recommend to thoroughly review the text to improve clarity and consistency.

We have revised the manuscript extensively to address this issue

Methods: "Quality of genotyping from the shared SNPs in both arrays was assessed before imputation separately by removing SNPs as follows" Does that mean that only the overlapping SNPs were used before imputation? What was the number of overlapping SNPs between these two panels? Where all the samples imputed together? Or by cohort, or by panel?

We have included this information in the methods: Genotype imputation (page 15).

Minor comments:

- The naming of the suppl are difficult to follow, as the name does not correspond to the suppl table number. Also, the tables do not have caption, which would be appreciated. So I still do not know which table is which.

We apologize for the confusion. We have now added full descriptions of the table in a supplemental note and corrected the numbering of the tables.

- I miss in the discussion a section that tells out how much improvement was gained by augmenting the sample size. How much larger was the number of eQTLs identified? Was there an improvement in identifying eQTLs driven by low-frequency variants? How much did the number of candidate effector genes improve?

We have extended our discussion to add this context and information (Page 12, 1st paragraph of Discussion).

Reviewer #3 (Remarks to the Author):

This manuscript describes a large eQTL dataset generated in pancreatic islets, the first such dataset I think that exists. It will be a useful resource for many researchers and the description of it in the manuscript is clear. The demonstration that cell type specificity of effects is important, and fits with other data in this area, and the match of cell type eQTL to previously assigned GWAS categories is a nice result.

I have only minor comments:

1) I was not familiar with the notion of p value enrichment analysis, nor the technique to perform it (p4). I had to go read the referenced paper, and I think it would be helpful if the authors were to include a brief outline of what the method intends to achieve, and how, before presenting their results.

We have added text clarifying the nature of the analysis in the results before presenting the results (Page 3, last paragraph).

2) On page 5/6 eSNPs within stretch enhancers are described as having "smaller effects" (last line on p5). Then (top of p6) such eSNPs are described as requiring larger sample sizes for "equivalent effect size". This is confusing and needs clarifying. Presumably the two "effect size" used mean different things here (eg fold change vs variance explained?).

We thank the reviewer for suggesting this clarification. We edited the text (pages 5/6) to address this. Briefly, we note the inverse relationship between the eQTL effect size and the number of samples

required to identify a significant association. Below we include a reviewer figure that shows the result of a power analysis to illustrate this point. As an example, if we hold power constant at 80% and MAF constant at 0.2, we can see that a SNP with an eQTL effect size of 0.12 can be detected with ~100 samples, whereas a SNP with eQTL effect size of 0.08 would take >250 samples to detect. We have modified the original text that was on pages 5 and 6 to clarify our initially confusing phrasing and we thank the reviewer for pointing this out. The figure below was generated using <https://github.com/sterding/powerEQTL>.

3) When this effect is discussed later (p12) it is suggested that this means that a GWAS causal variant sitting in an islet stretch enhancer could be misassigned to a non-islet expressed gene because its effect on islets differs from its effect on bystander genes in other tissues. I don't think this is shown - the effects of stretch enhancer SNPs to non stretch enhancer SNPs is compared within islets, but the effects of islet stretch enhancer SNPs on islet genes vs non-islet genes (where the stretch enhancer is not operating as a stretch enhancer??) are not compared. Either I have misunderstood this argument and it needs making clearer, or additional data are needed to support the claim.

We agree with the reviewer here that our analyses do not directly test if the SNPs in non stretch enhancer regions in non-islet tissues might be identified as eQTL for 'bystander' genes in other tissues. For this reason, we have removed this aspect from the discussion.

4) p9, discussing colocalisation of PDE8B, the authors describe the existence of two signals, which violates the assumption of the coloc approach. They have the data, and can condition on one of the two signals for each trait to test for colocalisation of the "other" signal (so 4 tests in total). If only summary data are available, COJO enables this.

For GWAS hits, the RTC approach does account for secondary signals, and we report colocalization results with two levels of evidence: 1) consensus between two methods, which we agree favors primary signals; and 2) colocalizing signals supported with just one method, which reports any possible secondary signals, as those are evaluated by RTC and included in the list. In addition, we have repeated the colocalization analysis using the latest and most up to date T2D GWAS study available (Mahajan, 2018). In this study only one GWAS association is reported for the ZBED3 locus. For this signal, we find again evidence for colocalization (COLOC = 0.99) between the eQTL for PDE8B (rs335628), and the lead GWAS variant (rs4457053). RTC again failed to report evidence for colocalization.

5) I would encourage the authors to complete the deposition of the data in EGA, and editors to confirm deposition is complete, because the sharing of this dataset will enable the widest utility for the work. (These authors have a previously very good record of sharing data - I'm saying this only because previous papers I have reviewed by other authors with "deposition in progress" have not always resulted in an actually deposited dataset).

We have now deposited all data with the following accession numbers for the different datasets: Genotype, technical and biological covariates, and sequence data have been deposited at the European Genome-phenome Archive (EGA; <https://www.ebi.ac.uk/ega/>) under the following accession numbers: EGAS00001003997; EGAS00001004042; EGAS00001004044; EGAS00001004056. Complete summary statistics for eQTL associations have been deposited and are accessible in the following link: <https://zenodo.org/record/3408356>

Reviewer comments, second version:

Reviewer #1 (Remarks to the Author):

The authors have probably done the best they could to reply to the reviewers. My only addition is that the authors should provide please a detailed comparison of the paper by Rai et al: <https://www.sciencedirect.com/science/article/pii/S2212877819309573>

What extra information/advantages are provided by the results under submission?

This comparison will be very valuable for the field.

Reviewer #4 (Remarks to the Author):

I think the authors have mostly addressed all the comments. But I have some minor concerns:

- Figure 2B p-values: the authors wrote in the figure legend “eQTL SNPs in ATAC-seq peaks in stretch enhancers have significantly lower effect sizes than SNPs in ATAC-seq peaks in active TSS and typical enhancer states”, but the p-value of differences in absolute effects sizes between eQTLs in stretch enhancers and typical enhancers, according to the figure included in the manuscript, is $p=0.13$. Of note, main figures that were provided separated from the manuscript show different p-values than the text -see Page 5- and merged figures: p (stretch enhancers vs typical enhancers) = 0.0298 (0.13 in manuscript text/figures), p (stretch enhancers vs active TSS)=0.0058 (0.0034 in manuscript text/figures), and p (active TSS vs typical enhancer)=0.6036 (0.88 in manuscript text/figures). Please clarify this.
- Overall results from Figure 2B and following discussion. The authors show in Figure 2B that eQTLs overlapping stretch enhancers had significant lower effect sizes than those in active TSS sites annotations. They linked these results to previous observations of regulatory elements showing robustness to regulatory variation. Stretch enhancers could encompass long stretches of DNA, parts of which do not include accessible chromatin regions that are likely to contain regulatory sequences, which could alter the distribution of eQTL effect sizes. The authors should assess if their conclusions hold true when analyzing eQTL effect sizes among other tissue-specific enhancer grouping definitions such as enhancer clusters or super-enhancers, or by integrating enhancer definitions using higher-resolution accessible chromatin such as those provided by Miguel-Escalada et al. 2019. This could provide further support to their notion of enhancer redundancy as the most plausible underlying cause of low eQTL effect sizes in these enhancer domains. Thus, this might require more systematic analyses or toning down the Discussion.
- Figure 3G: the EMSA figure does not clearly show which bands exhibit specific high binding affinity. Please explain which are interpreted as specific and which are not, and why.

Other comments:

- SuppTable1 corresponds to eQTLs from exon-level analysis and has 9,068 lines (9,069 with the header). However, in the manuscript, the authors reported 7,741 independent exon-QTLs. Please resolve this.
- Page 3, is “6p” correct? “set of 7.741 exone-level islet eQTLs overlapped eQTLs detected in 44 tissues ($n > 70$) version 6p of GTEx”.
- Page 5, “interactions between genotype and cellular fraction estimates, controlling for technical variables (Methods)”. I was not able to find this in the Methods section, I assumed that “technical variables” are the same used in the eQTL mapping but this could be easily clarified.
- Page 8. “We detected evidence for colocalization (using either method) for islet eQTLs at 46 GWAS loci (47 independent signals, Supp. File 1)”. SuppTable19 comprises all joint results for colocalization based on coloc and/or RTC and reports 53 loci (see also New_Loci_Index column). Are these the final results or did the authors applied additional filters? If not, this is not consistent with the text. In addition, the sheet’s name is not informative (the authors might double check the rest of excel supplementary tables).

- Page 11: “Three (rs7798124, rs7798360, rs7781710, Figure 3D, “Element 1”)”. Figure 3D shows normalized DGKB genes expression relative to allele dosage of lead eQTL. This could correspond to Figure 3E.
- REF 57 is now published, no longer a preprint in biorxiv:
<https://www.sciencedirect.com/science/article/pii/S0002929720300124>
- REF 81 is wrong see: “(!!! INVALID CITATION !!! 35)”.

Author rebuttal, second version:**Reviewer #1 (Remarks to the Author):**

The authors have probably done the best they could to reply to the reviewers. My only addition is that the authors should provide please a detailed comparison of the paper by Rai et al:

<https://www.sciencedirect.com/science/article/pii/S2212877819309573>

What extra information/advantages are provided by the results under submission?
This comparison will be very valuable for the field.

We thank the editor for this suggestion. The Rai et al paper¹ presented pancreatic islet single-cell-combinatorial-indexing ATAC-seq (sci-ATAC-seq) data that included chromatin accessibility profiles in islet alpha and beta cell clusters. In our manuscript, the islet eQTLs were identified from RNA-seq in bulk islets and compared to integrated ATAC-seq data across 33 bulk islet samples. For comparison with single nuclei chromatin profiles, we initially included beta cell ATAC-seq tracks in the current Fig. 4D. To expand on this, we now include two additional comparisons to the sci-ATAC data. First, we compute eQTL enrichments in the sets of peaks from the bulk islet and single nuclei islet alpha and beta cell clusters, which is represented as current Supplementary Figure 7F (also copied below). This new eQTL enrichment analysis is complementary and adds new information relative to the GWAS enrichment analysis we performed in the Rai et al. paper. Second, in the Supplementary Data 1 file, we now also indicate if the eQTL lead SNP overlaps ATAC-seq peaks in bulk islets, alpha, and beta cells.

Supplementary Figure 7F: eSNP fold enrichment in ATAC-seq peaks in islets and islet alpha and beta cells.

Reviewer #4 (Remarks to the Author):

I think the authors have mostly addressed all the comments. But I have some minor concerns:

- Figure 2B p-values: the authors wrote in the figure legend “eQTL SNPs in ATAC-seq peaks in stretch enhancers have significantly lower effect sizes than SNPs in ATAC-seq peaks in active TSS and typical enhancer states”, but the p-value of differences in absolute effects sizes between eQTLs in stretch enhancers and typical enhancers, according to the figure included in the manuscript, is $p=0.13$. Of note, main figures that were provided separated from the manuscript show different p-values than the text -see Page 5- and merged figures: p (stretch enhancers vs typical enhancers) = 0.0298 (0.13 in manuscript text/figures), p (stretch enhancers vs active TSS)=0.0058 (0.0034 in manuscript text/figures), and p (active TSS vs typical enhancer)=0.6036 (0.88 in manuscript text/figures). Please clarify this.

We apologize that different versions of figures were mistakenly uploaded. We have now updated the manuscript text and figure versions so that they are synced and consistent.

- Overall results from Figure 2B and following discussion. The authors show in Figure 2B

that eQTLs overlapping stretch enhancers had significant lower effect sizes than those in active TSS sites annotations. They linked these results to previous observations of regulatory elements showing robustness to regulatory variation. Stretch enhancers could encompass long stretches of DNA, parts of which do not include accessible chromatin regions that are likely to contain regulatory sequences, which could alter the distribution of eQTL effect sizes. The authors should assess if their conclusions hold true when analyzing eQTL effect sizes among other tissue-specific enhancer grouping definitions such as enhancer clusters or super-enhancers, or by integrating enhancer definitions using higher-resolution accessible chromatin such as those provided by Miguel-Escalada et al. 2019. This could provide further support to their notion of enhancer redundancy as the most plausible underlying cause of low eQTL effect sizes in these enhancer domains. Thus, this might require more systematic analyses or toning down the Discussion.

We thank the reviewer for these suggestions. We note our analyses focused on open chromatin, defined by islet ATAC-seq, within the chromatin state annotations, including stretch enhancers. We labeled the x-axis of Fig. 2B to indicate this. So, the effect size distributions we observe arise strictly from open chromatin regions and do not represent any DNA segments that may contain regulatory sequences in inaccessible chromatin. We believe this is comparable to the “enhancer definitions using higher-resolution accessible chromatin” suggestion made above.

We additionally performed analyses based on the datasets suggested above. Miguel-Escalada et al 2019² identified a ‘robust’ set of ATAC-seq peaks and partitioned these into groups such as active promoters, and three enhancer classes (Active Enhancer I, II and III). We also considered islet super enhancer segments overlapping islet robust ATAC-seq peaks presented in their work. We compared effect sizes of eSNPs occurring in these ATAC-seq peak annotations and observed that the eSNP effect sizes in accessible super enhancers are significantly lower than eSNP effect sizes in accessible active promoters ($P=0.041$), which is consistent with the trend observed with stretch enhancers in Fig. 2B. We did not observe significant differences with the three ATAC-seq peak enhancer classes. We have included these comparisons in Supplementary Figure 7E (also copied below).

Supplementary Figure 7E: Absolute effects sizes of eSNPs occurring in islet ATAC-seq ‘robust’ peak annotations such as active promoters, active enhancers I, II and III, and accessible super enhancers obtained from Miguel-Escalada et al 2019². The number of eSNP overlaps in each annotation are shown in parentheses. P values are from Wilcoxon rank sum tests.

- Figure 3G: the EMSA figure does not clearly show which bands exhibit specific high binding affinity. Please explain which are interpreted as specific and which are not, and why.

We thank the reviewer for pointing out the need for clarification here. In our EMSA experiment, proteins that bind to probes in a non-allele-specific manner may bind to sequences in the probe that do not contain the variant allele, or they may bind non-discriminately to oligonucleotides. We have modified Fig. 3G by adding filled arrowheads that point to allele-specific high binding affinity and open arrowheads that point to non-allele-specific binding. We further edited the figure legend and text to describe the arrowheads.

*Figure legend editions: F) Electrophoretic mobility shift assay (EMSA) for probes with risk and non-risk alleles at the four SNPs overlapping the regulatory element validated in (E) using nuclear extract from MIN6 cells. **Filled arrows, allele-specific binding; open arrows, non-allele-specific binding of proteins to probes.***

*Main text editions: (Figure 4F, **filled arrows**).*

Note: after the reviews, Figure 3 is now Figure 4.

Other comments:

- SuppTable1 corresponds to eQTLs from exon-level analysis and has 9,068 lines (9,069 with the header). However, in the manuscript, the authors reported 7,741 independent exon-QTLs. Please resolve this.

We apologize, there was an error during the formatting of the final file. The table has now been corrected.

- Page 3, is “6p” correct? “set of 7.741 exone-level islet eQTLs overlapped eQTLs detected in 44 tissues (n > 70) version 6p of GTEx”.

Yes, this is correct. After the release of version 6 for internal analyses, the GTEx consortia modified the quantification pipeline to correct for some issues that were identified at the time. Since both versions are still available for some researchers, we believe it is important to specify that our analyses were done in the final released version. This version is called 6p in the GTEx internal documentation.

- Page 5, “interactions between genotype and cellular fraction estimates, controlling for technical variables (Methods)”. I was not able to find this in the Methods section, I assumed that “technical variables” are the same used in the eQTL mapping but this could be easily clarified.

We thank the reviewer for identifying this oversight from our part. The methods of that analysis was entirely missing. These are now fully described in the methods.

- Page 8. “We detected evidence for colocalization (using either method) for islet eQTLs at 46 GWAS loci (47 independent signals, Supp. File 1)”. SuppTable19 comprises all joint results for colocalization based on coloc and/or RTC and reports 53 loci (see also New_Loci_Index column). Are these the final results or did the authors applied additional filters? If not, this is not consistent with the text. In addition, the sheet’s name is not informative (the authors might double check the rest of excel supplementary tables).

We apologize for the confusion, the table included 3 loci with discordant results between methods that were discussed in an earlier version of the manuscript. Those have been now removed from the table. We have also revised the file names, we hope they are more informative now.

- Page 11: “Three (rs7798124, rs7798360, rs7781710, Figure 3D, “Element 1”)”. Figure 3D shows normalized DGKB genes expression relative to allele dosage of lead eQTL. This could correspond to Figure 3E.

We have now corrected the text. These now belong to Figure 4.

- REF 57 is now published, no longer a preprint in biorxiv:

<https://www.sciencedirect.com/science/article/pii/S0002929720300124>

The reference has now been updated.

- REF 81 is wrong see: “(!!! INVALID CITATION !!! 35)”.

We have now corrected the formatting error.

Reviewer comments, third version:

Author rebuttal, third version: